# Fluorescence from a single-molecule probe directly attached to a plasmonic STM tip

Niklas Friedrich [1,10] ✉, Anna Rosławska [2], Xabier Arrieta[3], Katharina Kaiser [2,4], Michelangelo Romeo [2], Eric Le Moal [5], Fabrice Scheurer[2], Javier Aizpurua [6,7,8], Andrei G. Borisov [5], Tomáš Neuman [5,9] ✉ & Guillaume Schull [2] ✉

The scanning tunneling microscope (STM) provides access to atomic-scale properties of a conductive sample. While single-molecule tip functionalization has become a standard procedure, fluorescent molecular probes remained absent from the available tool set. Here, the plasmonic tip of an STM is functionalized with a single fluorescent molecule and is scanned on a plasmonic substrate. The tunneling current flowing through the tip-molecule-substrate junction generates a narrow-line emission of light corresponding to the fluorescence of the negatively charged molecule suspended at the apex of the tip, i.e., the emission of the excited molecular anion. The fluorescence of this molecular probe is recorded for tip-substrate nanocavities featuring different plasmonic resonances, for different tip-substrate distances and applied bias voltages, and on different substrates. We demonstrate that the width of the emission peak can be used as a probe of the exciton-plasmon coupling strength and that the energy of the emitted photons is governed by the molecule interactions with its environment. Additionally, we theoretically elucidate why the direct contact of the suspended molecule with the metallic tip does not totally quench the radiative emission of the molecule.

Scanning probe microscopes (SPM) have revolutionized our perception of the atomic-scale world, providing topographic, electronic, magnetic, optical or mechanical information of a surface with sub-nanometre spatial resolution. To address some specific properties of the probed sample, it proved advantageous to modify the chemical nature of the extremity of the scanning probe tip. Molecule-functionalized tips have played here a key role, enabling unprecedented resolution over the skeleton of molecules[1,2] or addressing electrostatic[3–6], magnetic[7,8] and transport properties[9–11]. In parallel, tip-induced electroluminescence[12–17], photoluminescence[18–20] or Raman spectroscopies[21,22] have reached sub-nanometer spatial resolution by making use of the large confinement of electromagnetic fields at the end of scanning tunneling microscope (STM) tips made of plasmonic materials. These techniques have made it possible to measure the fluorescence and scattering properties of individual molecules lying flat on thin decoupling layers with sub-molecular precision. Transferring a fluorescent molecule to the tip of an SPM may allow sensing and mapping the local electromagnetic field of a sample with the same spatial resolution. This has been attempted with terylene molecules embedded in micrometer-sized crystals[23], semiconductor quantum

[1]CIC nanoGUNE-BRTA, Donostia-San Sebastián, Spain. [2]Université de Strasbourg, CNRS, IPCMS, Strasbourg, France. [3]Materials Physics Center, CSIC-UPV/EHU, Donostia-San Sebastián, Spain. [4]IV. Physical Institute - Solids and Nanostructures, Georg-August-Universität Göttingen, Göttingen, Germany. [5]Université Paris-Saclay, CNRS, Institut des Sciences Moléculaires d'Orsay, Orsay, France. [6]Donostia International Physics Center, Donostia-San Sebastián, Spain. [7]Department of Electricity and Electronics, FCT-ZTF, UPV/EHU, Leioa, Spain. [8]IKERBASQUE, Basque Foundation for Science, Bilbao, Spain. [9]Institute of Physics, Czech Academy of Sciences, Prague, Czech Republic. [10]Present address: Institute of Experimental and Applied Physics, University of Regensburg, Regensburg, Germany. ✉e-mail: niklas.friedrich@ur.de; neuman@fzu.cz; guillaume.schull@ipcms.unistra.fr

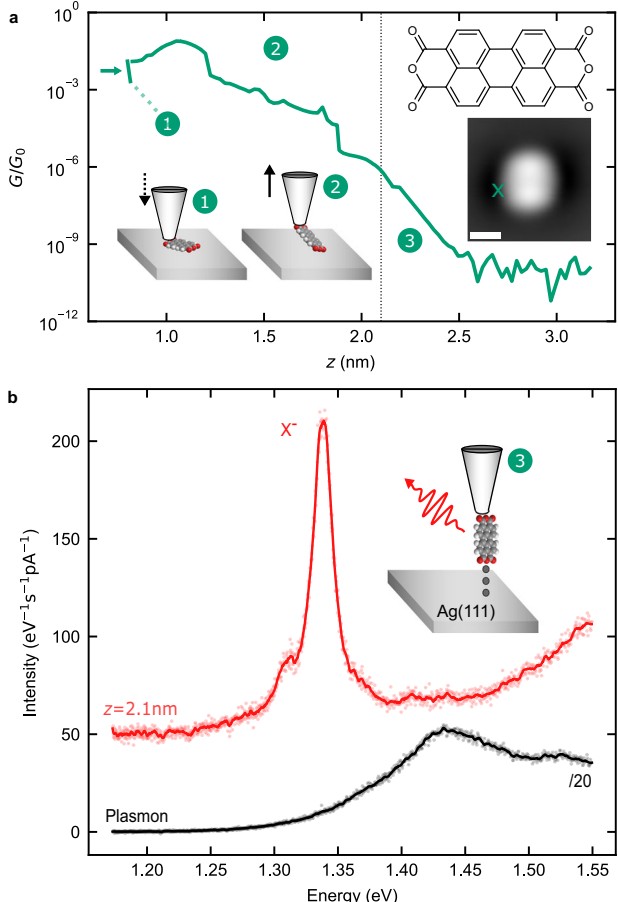

**Fig. 1 | Creation of a fluorescent molecular STM probe. a** Conductance *G* as a function of tip-substrate distance *z* recorded during the PTCDA attachment procedure in units of the conductance quantum $G_0$. The tip is approached towards the molecule with $V = 20$ mV (green dotted line, (1)) until a contact forms, indicated by the sudden increase of conductance (green arrow). Upon retracting the tip, the molecule is first cleaved from the substrate (green solid line, (2)) and finally, at 2.1 nm completely detached from the substrate ((3), loss of substrate-molecule contact indicated by the black vertical line). At 2.15 nm, the bias voltage is increased from 20 mV to 2.5 V. The inset is an STM constant current image ($I = 30$ pA, $V = -300$ mV, scale bar is 1 nm) of the molecule before lifting, as well as its Lewis structure. The position where the tip is approached is indicated by a green cross. **b** STML spectra of a metallic (black) and a PTCDA tip (red) in front of a bare Ag(111) surface. The peak at ≈1.34 eV is associated to the trion fluorescence ($X^-$) of the suspended PTCDA. $X^-$ acquisition parameters: $V = 2.5$ V, $I = 70$ pA, $t = 6$ min. Plasmon acquisition parameters: $V = 2.5$ V, $I = 50$ pA, $t = 2$ min. Source data are provided as a Source Data file.

performing a theoretical analysis, we show that in this configuration the spatial overlap between the molecular and tip electronic orbitals is much smaller than in geometries where the molecule would lie flat on the substrate. This leads to a reduced charge-transfer rate between the molecule and the metal, thus preserving the molecular fluorescence properties. As the PTCDA molecule is hanging approximately aligned with the tip axis in the STM junction, the molecular transition dipole is close to collinear with the plasmonic electric field in the gap, a situation that strongly contrasts with the on-surface flat-lying configuration of the molecules studied in usual tip-induced fluorescence measurements[19]. This collinear geometry leads to a large increase of the coupling strength between the molecular exciton and the tip-sample plasmonic cavity by up to two orders of magnitude compared to the perpendicular (flat-lying) configuration. Eventually, the spectral characteristics of the PTCDA tips are investigated as a function of the plasmonic response of the tip-sample cavity, the bias voltage, the tip-sample distance, and the lateral distance from an atomic-scale defect of the surface revealing that the emission of PTCDA tips can be used as a probe of the local electromagnetic and electrostatic fields.

## Results

### Creating a fluorescent STM tip

The procedure for making fluorescent PTCDA tips in a low-temperature, ultrahigh vacuum STM is schematized in Fig. 1 (see Methods for details of the sample preparation). A silver-covered tungsten tip is approached (dotted line in Fig. 1a) to the oxygen atoms located at the extremity of a PTCDA molecule deposited on Ag(111) until a contact is reached (current jump indicated by the green arrow in Fig. 1a). At this contact point, the last tip atom is expected to be at $z = 0.8$ nm above the metallic surface[29]. The tip is then retracted together with the attached molecule at its extremity. During this procedure the electrical current traversing the junction is monitored (solid green line Fig. 1a). The presence of the molecule at the tip is confirmed by the much larger conductance values recorded during retraction ((2) Fig. 1a).

Eventually, the complete detachment of the molecule from the surface is identified ((3) Fig. 1a) at $z > 2.1$ nm by a strong slope change of the decaying conductance with distance. As discussed in refs. 30,31, the PTCDA molecule adopts an unexpected up-standing configuration at the tip apex thanks to stabilizing electrostatic dipole forces that prevent the molecule from toppling over onto the tip shaft. Besides, it has been demonstrated that for low positive voltage tunneling conditions ($V < 3$ V), the attached molecule is negatively charged (PTCDA$^-$)[27,28,32–34], a state that reflects the electron acceptor character of the molecule. In Fig. 1b we report STM-induced luminescence (STML) spectra acquired at $V = 2.5$ V, first with a clean metal tip, revealing a broad feature characteristic of plasmonic emission (black spectrum), and then with a fully detached ($z = 2.1$ nm) PTCDA tip on top of the bare silver surface (red spectrum). In contrast to the spectrally broad plasmonic emission observed when the clean tip is used, the STML spectrum measured using the PTCDA tip exhibits a much sharper emission peak, which is centered at a photon energy of 1.34 eV. In agreement with a recent report[17], we assign this peak to the luminescence of negatively charged PTCDA (later referred to as negative trion $X^-$). This result provides experimental evidence that the luminescence properties of the molecule suspended at the tip are preserved, despite the direct molecule-metal contact. The PTCDA tip can therefore be considered as a fluorescent molecular probe, whose spectral emission properties measured in the far field can be used to sense its local environment.

### Fluorescence response to the local plasmonic environment

As a first step to characterize this fluorescent probe, we compare spectra (Fig. 2a) successively obtained with a PTCDA tip located above the bare Ag(111) (red spectrum), above a two atomic layer-thick (2 ML) NaCl island grown on Ag(111) (blue spectrum) and with the exact same

dots (SCQD) fixed on the tip of an optical fiber[24], and nitrogen-vacancy centers in diamond nanocrystals attached to atomic force microscopy tips[25]. Whereas these nearfield probes have been used to sense electrostatic, electromagnetic and magnetic fields, the spatial resolution they provide remains limited by the (large) size of the emitter (SCQD) or of the nanocrystal into which the emitter is embedded. Hence, SPM tips preserving the fluorescence properties of a single molecule directly attached to the tip apex are highly desirable, but have not been reported so far. This is presumably because direct contact between the molecule and the metal generally causes total quenching of the molecule fluorescence, due to ultrafast electron and energy transfer from the excited molecule to the metal[26].

Here we build on recent works demonstrating that a single 3,4,9,10-perylenetetracarboxylic-dianhydride (PTCDA) molecule can be suspended at the apex of a metallic STM tip[27,28], and report on the electrically-induced fluorescence of this functionalized PTCDA tip. By

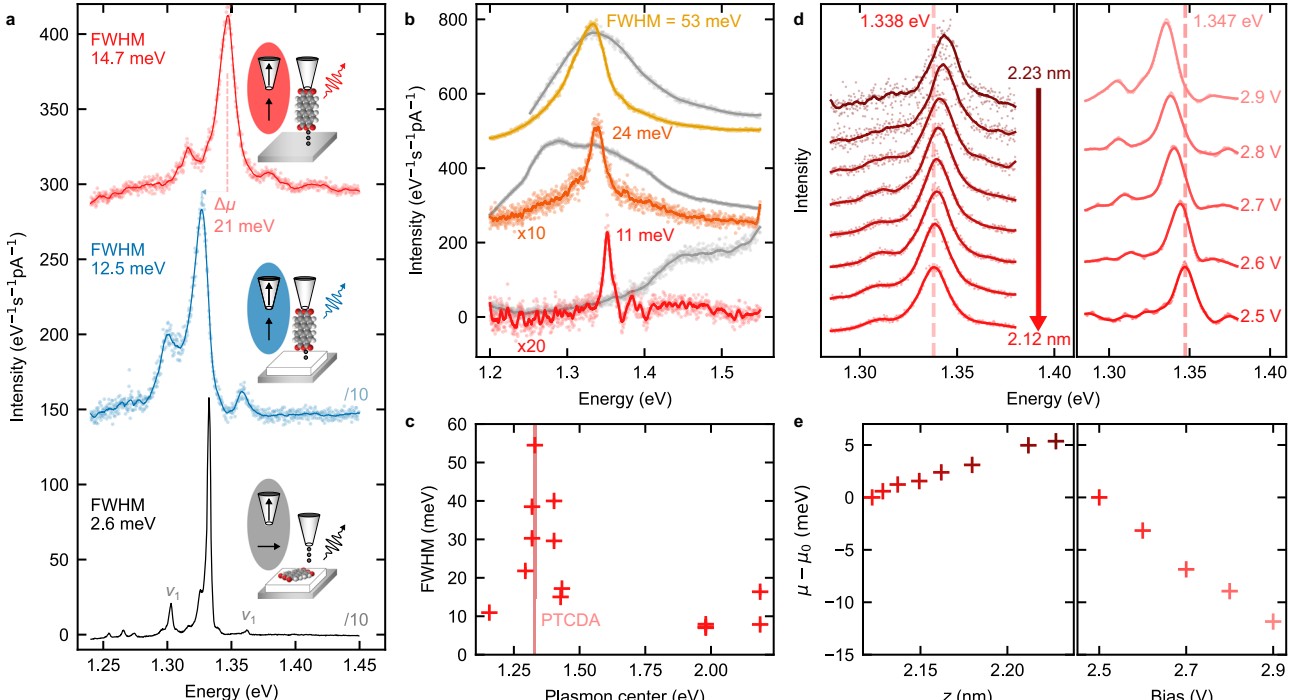

**Fig. 2 | Response of fluorescent probe to electrodynamic and electrostatic environment. a** STML spectra of a PTCDA tip in front of Ag(111) (red) and of 2 ML NaCl (blue), and of the flat-lying PTCDA on 3 ML NaCl excited with a metal tip (black). This data set was recorded with the same molecule. The plasmonic response of the tip (without molecule attached) is presented in Supplementary Fig. 1 showing a very broad emission around 1.4–1.5 eV. **b** X⁻ emission spectra (yellow, orange and red) for three different PTCDA tips above Ag(111) and the associated plasmonic spectra (in gray) recorded prior to the molecular functionalization of the tip. The FWHM of the X⁻ peak increases for resonant plasmon-trion conditions. **c** FWHM of X⁻ as a function of the energy of the plasmon peak maximum. These data were recorded at an identical set point ($I = 30$ pA, $V = 2.5$ V) and stem from 13 macro- or microscopically different tips. **d** X⁻ emission spectra for varying PTCDA-tip-to-molecule surface separation (left, $V = 2.5$ V) and for varying bias voltage (right, $z = 2.2$ nm). A red shift of the central emission is observed upon reducing $z$ or increasing $V$. **e** Relative shift of the central emission line $\mu$ extracted from the data in (**d**), taking the reference energy $\mu_0$ from the bottom spectrum in (**d**). Color coding corresponds to the spectra in (**d**). Spectra in (**a**, **b**, **d**) are offset vertically for easier comparison. Both $\mu$ and the FWHM of the spectra were extracted by fitting a Lorentzian function to the data, accounting for the vibronic satellites with Lorentzian functions when necessary. Acquisition parameters in (**a**) from top to bottom: $V = 2.5, 2.5, -2.5$ V, $I = 60, 50, 60$ pA, $t = 6, 6, 30$ min. Spectra acquisition parameters for (**b**–**d**) are detailed in Supplementary Note 1.4. Source data are provided as a Source Data file.

molecule adsorbed flat on 3 ML NaCl but excited using a bare metal tip (black spectrum). For this last spectrum, the molecule was released from the tip by applying a low-voltage pulse on top of NaCl. All spectra exhibit an intense X⁻ emission line together with two vibronic satellites ($v_1$), located at ≈30 meV on each side of the X⁻ line, that occur from an in-plane breathing mode of the molecule[35]. The satellite peak on the high-energy side of the X⁻ line is assigned to hot luminescence[36] (see also additional analysis in Supplementary Note. 2.6). When the PTCDA tip is used, we observe that the STML spectrum measured on top of 2 ML NaCl-covered Ag(111) is rigidly redshifted by $\Delta\mu = 21$ meV relative to the spectrum measured on bare Ag(111). This shift suggests that the emission properties of the tip-suspended molecule are dependent on its electrostatic environment (Stark shift), which is modified by the presence of the NaCl island, a conclusion that will be further discussed below.

Moreover, the full width at half maximum (FWHM) of the emission peak is one order of magnitude larger in the spectra measured using the PTCDA tip (FWHM > 12 meV) than in the spectra measured using the clean tip on the PTCDA molecule lying flat on the NaCl (FWHM ≈ 2.6 meV). The width of molecular emission lines in STML experiments may have different origins and may be due to dephasing induced by the coupling of the trion with its electrostatic and/or phononic environments, non-radiative decay paths, the superposition of multiple peaks[19] or a strongly shortened trion fluorescence lifetime compared to the free molecule because of the particular electromagnetic environment of the STM junction (Purcell effect). When the latter

dominates, the FWHM is determined by the trion-plasmon coupling strength.

To identify the respective role of dephasing and lifetime-shortening on the emission linewidth of the PTCDA tip, we investigate the FWHM of the X⁻ line as a function of the energy of the gap plasmon of the tip-sample junction. It is well known that the plasmonic response of the junction strongly depends on the nanoscale structure of the tip apex, a parameter that can be tuned by controlled indentations of the tip in the bare metal surface[37]. In Fig. 2b, we show the STML spectra of three different PTCDA tips above bare Ag(111) together with the plasmonic response of the tip-sample junction recorded before functionalization with the PTCDA molecule. The plasmon peak is either strongly blue-detuned (bottom spectrum), weakly red-detuned (middle spectrum), or in-resonance (top spectrum) with the X⁻ line. Here, the smaller the frequency mismatch between the plasmonic resonance and the trion emission, the broader the FWHM of the X⁻ emission. This correlation is better evidenced in Fig. 2c where the FWHM of the X⁻ line is plotted as a function of the energy of the plasmon resonance maximum for 13 different tips. This plot also reveals that the FWHM does not reach values lower than 7 meV, even for non-resonant conditions. This suggests that the line width is in this case limited by dephasing or non-radiative decay paths that do not involve coupling to plasmons. In contrast, the FWHM increase observed at resonance indicates a reduction of the excited state lifetime due to the stronger coupling of the trion with the gap plasmons. The FWHM then reaches values as high as 53 meV, approximately

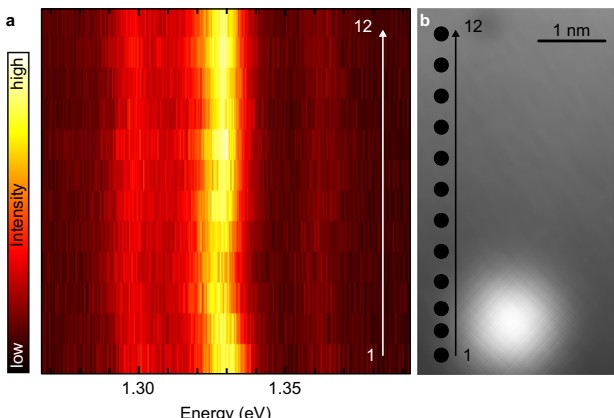

**Fig. 3 | Sensitivity to atomic defects. a** Constant height STML spectra ($V = 2.3$ V, $t = 1$ min per spectrum) of a PTCDA tip acquired at different lateral distances (see dots in **b**) from a defect on 2 ML NaCl. **b** STM image ($I = 5$ pA, $V = -0.5$ V) of the defect acquired with a metal tip. Source data are provided as a Source Data file.

corresponding to a fluorescence lifetime of 75 fs. The trion-plasmon coupling is here still below the threshold of strong coupling[38]. As a figure of merit to quantify the regime of plasmon-trion coupling, we use the ratio between the experimental plasmon-trion coupling $g$ (i.e., the Jaynes-Cummings coupling strength as defined in ref. 16) and the experimental width of the plasmon $\kappa$. Assuming that the trion couples to a single plasmon mode as described in ref. 16, we estimate $g$ for the resonant case (top spectrum Fig. 2b) from the plasmon-induced width $\hbar\gamma_{pl} \approx 46$ meV (assuming that $\approx 7$ meV of the total $\approx 53$ meV broadening accounts for other mechanisms) and $\hbar\kappa \approx 100$ meV ($\hbar$ is the reduced Planck constant) as $\hbar g = \hbar\sqrt{\gamma_{pl}\kappa}/2 \approx 34$ meV. Since $2g$ is smaller than $\kappa$ we conclude that the plasmonic losses still dominate over the plasmon-trion coupling and we therefore observe a weak trion-plasmon coupling. For flat PTCDA adsorbed on 4 ML NaCl, a FWHM as low as 0.5 meV could be measured (see Supplementary Note 1.2), setting a lower limit for the trion lifetime of 8 ps when the tip and molecular dipoles are orthogonal to each other (compare sketches in Fig. 2a).

Overall, our results show up to two orders of magnitude shorter excited state lifetime of the PTCDA trion when the transition dipole of the molecule is parallel to the electric field of the picocavity plasmons than when it is orthogonal to it, i.e., the configuration used in previous STML studies. Furthermore, the coupling strength is maximum when the plasmon is tuned in resonance with the $X^-$ line. This also indicates that the FWHM of $X^-$ can be used to probe variations of the local density of photonic states at metallic surfaces, e.g., at the surface of plasmonic nanostructures[23,39].

### Fluorescence response to the electrostatic environment
In principle, not only the FWHM, but also the energy position of the $X^-$ line can be used for probing the environment of the molecular tip. In previous reports, emission peak positions have been shown to vary as a function of both the local electromagnetic field, an effect known as photonic Lamb shift[40], and the electrostatic field where the spectral shift is described in terms of a Stark effect[16]. In Fig. 2d, we monitor the $X^-$ line as a function of the PTCDA-tip-to-Ag(111) distance (at constant voltage) and as a function of the voltage (at constant tip-sample distance). The FWHM of the line ($\approx 22$ meV) remains constant for the two sets, indicating that the trion-plasmon coupling does not significantly vary over this range of distances or voltages. In contrast, the $X^-$ line experiences a shift to lower energies with tip approach and with increasing voltages (similar data are provided in Supplementary Note 1.3 for a PTCDA tip located on top of 2 ML NaCl). The voltage-dependent shift of the peak at constant tip-molecule height can be

assigned to the sole Stark effect due to the static field $E_V$ in the junction $\mu - \mu_0 = -c_S E_V$ (with $c_S$ being a constant), as the plasmon-exciton coupling (responsible for Lamb shifts) remains constant. Figure 2e reveals that this line-shift evolves essentially linearly with distance $z$, at a rate of $\approx 50$ meV nm$^{-1}$, and with voltage, at a rate of $\approx 30$ meV V$^{-1}$. Considering that $E_V = V/z$ in V nm$^{-1}$, these values are consistent with each other yielding $c_S \approx 0.06$ e nm. These findings are about one order of magnitude larger than the rates reported for phthalocyanine molecules lying flat on NaCl and addressed by a metal tip[16]. In this case, the line shifts caused by changes in the tip-sample distance and voltage were dominated by the Stark effect due to the electrostatic field in the STM junction. The same conclusion can be drawn for the 21 meV shift observed when the PTCDA tip is moved on top of 2 ML NaCl (Fig. 2a), which can be assigned to changes of the local work function of the substrate. Indeed, covering the Ag(111) surface with NaCl leads to a work function decrease by $\approx 1$ eV[41], which is equivalent to adding $\approx 1$ V of external bias. By extrapolating the experimental values obtained in Fig. 2e, we conclude that the exciton line should redshift by $\approx 30$ meV on NaCl, consistent with the observed $\approx 21$ meV shift. For completeness, in Supplementary Note 2.9 we discuss the possible role of other effects including the static and dynamical (Lamb shift) screening by the dielectric on the line shift. We conclude that the Stark shift due to the change in the substrate work function is one order of magnitude stronger than the other effects considered. Moreover, the Lamb shift would have the qualitatively opposite effect of blue shifting the resonance of the molecule on NaCl. Overall, this suggests that the peak shift observed in our data is dominated by the Stark effect.

To further illustrate the sensitivity of the PTCDA-probe to changes in its local environment, we recorded STML spectra for each pixel of a line-scan for a PTCDA tip that is laterally moved toward an atomic-scale defect located on a NaCl bilayer (Fig. 3). The spectra reveal a small, but observable, shift ($\approx 3$ meV) of the spectral line in the proximity of the defect. Together with the data of Fig. 2a, d, e, this demonstrates that one can use the spectral shift of the PTCDA tip emission line as a probe of the local electrostatic field.

## Discussion
To explain the unexpected observation of bright fluorescence from a molecule that is in direct contact with a metallic electrode, we perform a series of quantum chemical and quasi-electrostatic calculations. We develop a theoretical description that compares (i) the interaction between the molecular exciton and the tip/substrate plasmons as discussed in detail in ref. 16, and (ii) the charge transfer between the molecule and the tip, which is expected to strongly contribute to the quenching of the molecule's photon emission. Our model is thus able to capture the most relevant physical processes providing order-of-magnitude estimates of the photon emission efficiency.

We describe the charge transfer in an effective single-particle picture in which we estimate the charge transfer rate of several molecular orbitals into the metal surface using the real-time wave-packet propagation (WPP) technique[42] (see Supplementary Note 2.3 for details). Since our main goal here is to confidently explain why the radiative decay is not quenched by the electron transfer from the excited molecule into the metal, we consider adsorption geometries that maximize the electron coupling with the surface. The molecule is assumed to be positioned above a planar metal surface and the distance $a$ between the molecule and the image plane of the surface is varied. We consider two different configurations: the molecule lying flat on the surface (Fig. 4a) and the molecule in the lifted configuration with its plane perpendicular to the surface (Fig. 4b). In the latter case, $a$ describes the distance between the image plane and the bottom oxygen atoms of the molecule. We did not consider the finite curvature of the tip (see further discussion in Supplementary Note 2.5) or the geometry where the molecule is attached to two silver adatoms[30]. Indeed both situations a priori increase the overall distance from the

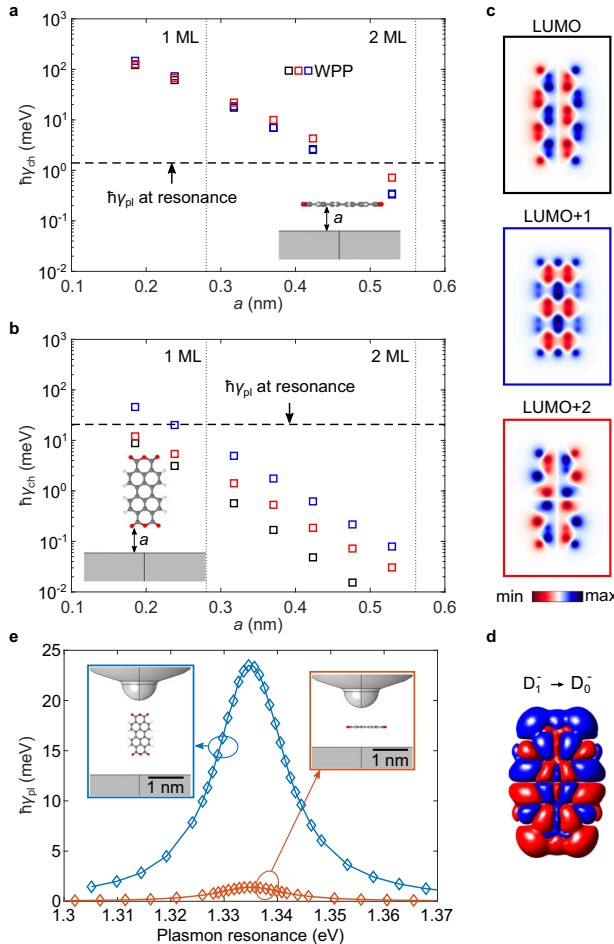

**Fig. 4 | Comparison of decay pathways.** Calculated charge transfer rates $\gamma_{ch}$ for (**a**) the molecule lying flat and (**b**) standing upright on a semi-infinite jellium interface as a function of the distance $a$ of the molecule from the jellium edge. The rates were calculated using the real-time wavepacket propagation method (WPP, squares) for orbitals shown in (**c**). The color of the frames in (**c**) corresponds to the color coding in (**a**, **b**). The dashed line in (**a**, **b**) corresponds to the maximum plasmon-induced decay rate $\gamma_{pl}$ shown in (**e**) for the respective configurations. **c** Molecular orbitals LUMO, LUMO+1, and LUMO+2 calculated using the WPP approach for an isolated molecule (see Supplementary Eq. 3). The orbitals are evaluated $1a_0$ ($a_0$ being the Bohr radius) above the plane of the molecule. **d** Transition charge density of the $D_1^- \rightarrow D_0^-$ transition calculated by TDDFT using Gaussian 16 rev. C.01[50]. **e** Plasmon-induced exciton decay rate as a function of the plasmonic resonance energy tuned by varying the length of the model plasmonic cavity (see Supplementary Fig. 5 for details of the geometry) calculated for the molecule in the upright configuration (blue) and the flat configuration (orange). The molecule is positioned in the middle of the tip-substrate gap which is 2.1 nm for the upright configuration and 1.1 nm for the flat one. Source data are provided as a Source Data file.

surface and reduce the electron transfer rate. The charge-transfer rate $\gamma_{ch}$ is shown in Fig. 4a (Fig. 4b) for the flat-lying (lifted) configuration for the LUMO (lowest unoccupied molecular orbital) (black), LUMO+1 (blue), and LUMO+2 (red) as squares. The orbital labeling refers to the electronic configuration of the neutral molecule, i.e., the LUMO is split into the singly occupied and unoccupied orbitals in the negative ground state $D_0^-$. We note that both LUMO and LUMO+2 have a significant contribution to the electronic configuration of $D_1^-$ (see Supplementary Notes 2.1 and 2.3 for details). The corresponding orbitals are shown in Fig. 4c. Figure 4b (Fig. 4a) demonstrates for all relevant orbitals that at a distance of about 0.2 nm - roughly corresponding to the distance between the atoms binding the tip and the molecule[30,31] -

the charge-transfer rate reaches up to ≈10 meV (≈100 meV) for the lifted (flat-lying) configuration. In other words, the lifted configuration leads to a ten-times weaker metal-molecule electronic coupling than for a flat-lying molecule.

To know if the charge transfer rate $\gamma_{ch}$ is fast enough to quench the trion emission, one should compare it with the radiative decay rate expressed by $\gamma_{pl}$. To calculate $\gamma_{pl}$, we first describe the electronic excitations in the negative PTCDA molecule using time-dependent density-functional theory (TDDFT) (see Supplementary Note 2.1 for details). We relax the molecular geometry in its first excited state $D_1^-$ and extract the molecule's transition charge density $\rho$ (Fig. 4d) - generalizing the concept of the transition dipole moment. We next calculate a quasi-static electric potential $\phi$ generated by the source charge density $\rho$ placed in the plasmonic environment formed by the tip and the substrate. We obtain the plasmon-induced broadening $\hbar\gamma_{pl}$ of the trion emission line as $\hbar\gamma_{pl} = 2\text{Im}\{\int \rho(\mathbf{r})\phi(\mathbf{r})d\mathbf{r}\}$[16] and show the result in Fig. 4e as a function of the plasmon resonance energy, which is artificially tuned by changing the tip geometry (see Supplementary Note 2.5). The molecule is considered to be parallel to the surface or in upright geometry in the center of the tip-substrate gap. As demonstrated in Fig. 4e, the resonance between the trionic state and the plasmonic mode is necessary to reach large trion broadening of up to 24 meV for the upright molecule, of the order of the experimental value reported in Fig. 2c (see Supplementary Note 2.7 for a detailed discussion). Our calculation thus supports the conclusion that the plasmon resonance is responsible for the observed broadening of the trion line. Besides, in the lifted configuration, the broadening is ≈15 times larger than the maximal plasmon-exciton coupling obtained for the flat-lying molecule (Fig. 4e). In other words, the fluorescence decay is ≈15 times faster for molecules aligned with the tip axis than for flat-lying molecules. The plasmon-induced trion decay rate $\gamma_{pl}$ in both configurations is represented by horizontal dashed lines in Fig. 4a, b. For the flat-lying case (Fig. 4a), in agreement with the analysis performed in ref. 42, the plasmon-induced trionic decay rate dominates over the charge-transfer rate $\gamma_{ch}$ only for $a \gtrsim 0.45$ nm, a molecule-substrate distance that is close to the one corresponding to 2 ML NaCl where an STM-induced fluorescence starts to be observed. In contrast, the plasmon-induced trion decay rate already dominates for $a \gtrsim 0.2$ nm for the lifted configuration (Fig. 4b). Computational details such as slight modification of the tip-protrusion geometry or the effect of a moderate tilt of the molecule with respect to the tip axis do not change our conclusions (see Supplementary Notes 2.5 and 2.7). Overall, these data show that the observation of fluorescence in the lifted configuration results from the combination of a reduced electronic coupling between molecule and electrode states with the enhanced electromagnetic coupling between the vertically polarized tip plasmons and the molecular excited states.

Next, we briefly discuss the mechanisms that are likely involved in the STM-induced excitation of the PTCDA molecule suspended on the tip. In STML the mechanism bringing the molecule to the excited state usually depends on a sequence of charge transfer events that are specific to the considered system[43,44]. Figure 5a shows a d$I$/d$V$ spectrum simultaneously recorded with the total intensity of the X$^-$ line emitted by the molecule. The d$I$/d$V$ curve shows a step-like increase of the bias voltage at ≈0.3 V and ≈2.3 V. The step at ≈2.3 V is accompanied by an equally smooth onset of photon emission. To explain this voltage-dependent behavior we propose a model of charge transfer events between the many-body electronic states of the molecule (see Fig. 5b). In the model we include electronic states of the neutral molecule (singlet ground state $S_0$ and a triplet state $T_1^0$) and two doublet states ($D_0^-$ and $D_1^-$) of the singly negatively charged molecule. The energies of the states displayed in the diagram are aligned with respect to the work function of the Ag tip which we estimate at $W_{Ag} \approx 4.5$ eV[45,46]. When a voltage $V$ is applied, the relative energy of the

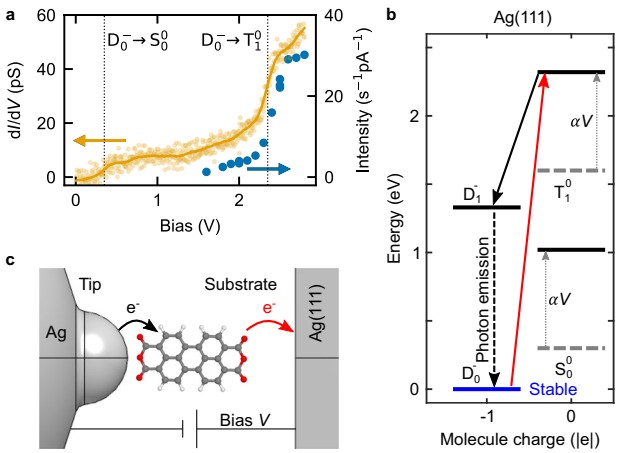

**Fig. 5 | Photon emission mechanism. a** d$I$/d$V$ (yellow) and photon intensity (blue dots) shown as a function of sample bias. **b** Tentative diagram of many-body electronic states of the molecule and the tip-mediated (black arrow) and substrate-mediated (red arrow) charge-transfer events followed by trion recombination (black dashed arrow). Due to the non-negligible voltage drop across the molecule, the relative energies of the states (aligned with respect to the Ag(111) work function) are dependent on the applied bias (see Supplementary Fig. 11) with $\alpha \approx 1/3$ (gray dotted arrows). The direct contact of the molecule with the tip stabilizes the molecule in the negative doublet state $D_0^-$, but when bias is applied, the molecule can be brought into the excited state $D_1^-$ by first releasing an electron into the substrate (red arrow) and then capturing an electron from the tip (black arrow) as discussed in the text. **c** Schematic depiction of the tip-molecule-substrate geometry where voltage is applied between the two electrodes and an electron is transferred from the tip to the molecule and subsequently from the molecule into the Ag(111) substrate. Source data are provided as a Source Data file.

different charge states shifts by $\alpha V$ as indicated by the gray dotted arrows. Here $\alpha$ indicates the fractional voltage drop experienced by the molecule with respect to the tip electrode. We envision the mechanism as follows. First, by applying a small bias of about $\approx 0.3$ V the singly negatively charged molecule can be transiently neutralized by tunneling to the substrate and brought into the $S_0^0$ state. For a voltage of $V \approx 2.3$ V the transition into the neutral triplet $T_1^0$ state is enabled (red arrows in Fig. 5b, c). Once in the neutral triplet state, the molecule can be rapidly charged from the tip (black arrows) and can end up in the excited doublet configuration ($D_1^-$). $D_1^-$ readily decays into the ground state either by non-radiatively exchanging an electron with the tip or by photon emission. We note that when the lifted molecule is brought above the NaCl/Ag(111) surface, the mechanism leading to light emission can differ from the present discussion (see Supplementary Note 2.8).

In conclusion, we demonstrated that a PTCDA molecule preserves its intrinsic emission properties even when it is directly attached to a plasmonic scanning probe tip. This is due to the relatively low spatial overlap between the tip and molecule electronic orbitals—which in turn leads to weak luminescence quenching by charge transfer—and by the strongly increased radiative decay probability (by up to 2 orders of magnitude compared to flat-lying PTCDA) of the molecular emitter fixed vertically at the apex of the plasmonic tip. This increased trion-plasmon interaction, however, remains insufficient to reach the strong coupling regime. While our data also demonstrate qualitatively that the fluorescent properties of the molecular probe are simultaneously sensitive to the dynamical electromagnetic, i.e., plasmonic, and electrostatic environment, additional work will be required for a full quantitative description of all effects. Eventually, the excitation mechanism by tunneling electrons has been discussed. This atomic-scale sensor could be used, in the near future, in resonant energy transfer microscopy experiments where it would provide atomic-scale lateral and axial precision.

## Methods

### Experimental setup
The STM data was acquired with a low temperature (5.5 K) Unisoku setup operating in ultrahigh vacuum and adapted to detect the light emitted at the tip-sample junction. The optical detection setup is composed of a spectrograph coupled to a CCD camera; a grating having a groove density of 300 lines per mm was used and provided a spectral resolution of $\approx 1$ nm for all the data presented in the paper with the exception of the data of Suppl. Fig. 3 where a grating with 1200 lines per mm was used. Details on the acquisition parameters for each spectrum are listed in Supplementary Note 1.4. Tungsten STM tips were introduced in the sample to cover them with silver so as to tune their plasmonic response.

### Sample preparation
The Ag(111) substrate was cleaned with successive Ar$^+$-ion sputtering and annealing cycles. Approximately 0.5 ML of NaCl were sublimated on Ag(111) kept at room temperature. The sample was then flash-annealed up to 370 K to obtain square domains of bi- and tri-layers of NaCl. PTCDA was evaporated in situ on the sample held at 5.5 K using a molecular beam evaporator ($T_{evap} = 613$ K), resulting in a sparse distribution of individual molecules.

### Data acquisition and processing
Differential conductance spectra were recorded with internal lock-in amplifier using a modulation amplitude of 20 mV at $f = 768$ Hz. Light and d$I$/d$V$ spectra were processed using custom Python scripts. The Python package lmfit[47] was used for all fitting procedures. STM images were processed using WSXM[48].

### Theoretical models
The detailed description of the employed theoretical models, including the precise software packages, is provided in Supplementary Notes 2.1–2.3 and 2.6.

### Reporting summary
Further information on research design is available in the Nature Portfolio Reporting Summary linked to this article.

## Data availability
The data that support the findings of this study are available from the corresponding authors upon request. Raw STM, STS, and STML data generated in this study are available from Zenodo[49] and provided as Suppl. Data 1. Processed STM, STS, and STML data, as well as data stemming from theoretical models are available from Zenodo[49]. Source data are provided with this paper.

## Code availability
The custom code used in this study is available from the corresponding authors upon request.

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

## Acknowledgements

We are grateful to H. Sumar, R. Baehr and V. Speisser for technical help in setting up the experiment. JA and XA acknowledge grant no. IT 1526-22 from the Basque Government for consolidated groups of the Basque University and grant PID2022-139579NB-I00 funded by MICIU/AEI/10.13039/501100011033 and by ERDF, EU, and XA acknowledges Spanish Ministerio de Ciencia, Innovación y Universidades for grant no. FPU21/02963. NF acknowledges funding from the Spanish government MICIU/AEI/10.13039/501100011033 through grants MAT2016-78293-C61 and PID2019-107338RB-C61, and by the European Union (EU) H2020 program through the FET Open project SPRING (grant agreement no. 863098). This work is supported by "Investissements d'Avenir" LabEx PALM (ANR-10-LABX-0039-PALM). TN acknowledges the Lumina Quaeruntur fellowship of the Czech Academy of Sciences.

Computational resources were supplied by the project "e-Infrastruktura CZ" (e-INFRA CZ LM2018140) supported by the Ministry of Education, Youth and Sports of the Czech Republic. This project has received funding from the European Research Council (ERC) under the European Union's Horizon 2020 research and innovation programme (grant agreement no. 771850). This work of the interdisciplinary Thematic Institute QMat, as part of the ITI 2021 2028 program of the University of Strasbourg, CNRS and Inserm, was supported by IdEx Unistra (ANR 10 IDEX 0002), as well as by SFRI STRAT'US project (ANR 20 SFRI 0012) and EUR QMAT ANR-17-EURE-0024 under the framework of the French investments for the Future Program. This work is supported by the Agence Nationale de la Recherche (ANR) under contract no. ANR-22-CE09-0008.

## Author contributions

G.S. conceived and designed the experiment. M.R. developed specialized software for the data acquisition. N.F., A.R., K.K., and E.L.M. performed the STML experiments. N.F. and K.K. analyzed the data. X.A., J.A., A.G.B., and T.N. developed the theoretical models and performed the numerical simulations. F.S. and all authors contributed to the discussion of the results. N.F., T.N., and G.S. wrote the manuscript with input from all authors.

## Competing interests

The authors declare no competing interests.
