## [Transparent Peer Review file · Nature Communications]

Fluorescence from a single-molecule probe directly attached to a plasmonic STM tip

Corresponding Author: Dr Niklas Friedrich

Version 0:

Reviewer comments:

Reviewer #1

(Remarks to the Author)

In this work the authors demonstrate fluorescence from a single molecule bound to a STM tip. Normally, the fluorescence of molecules on metal surfaces are quenched as the molecule tends to lie flat on the surface and thus have significant interactions with the surface. Here, the authors avoid this by using a PTCDA molecule which has previously been demonstrated to adopt a standing-up configuration on the tip due to electrostatic forces. They further demonstrate that the FWHM and fluorescence energy is sensitive to its local environment and thus could be used as a probe of the local electrostatic environment. A theoretical model explaining the results, in particular, why the fluorescence is not quenched is presented. Overall, this is a high quality work that demonstrates an important advance in the field and thus is well suited for Nature Communications. Below I have some technical questions that the authors should address before publication.

- It is assumed that the PTCDA molecule remain standing up with the transition dipole movement aligned with the near-field in the junction. While this is reasonable it would be important to characterize/estimate the influence of this assumption. Depending on the local environment one could expect that the molecule binds with different angles if the binding strength is not too large. This would be particularly important if this will be used as a probe of the local environment at envision by the authors. Could the authors provide an estimate of the binding strength and the influence of the alignment on the quenching?

- In Figure 2a the authors shows a shifts of 21 meV for the emission as the tip is brought closer to the NaCl substrate. However, the energy is lower than that of the molecule lying flat on the NaCl substrate and thus should have a larger perturbation. Why is this?

- The two satellite peaks (ν_{-1}) is assigned as coupling to molecular vibrations. The authors could easily assign this by calculating the vibronic effect using the TDDFT module. In the simulations the main peak is assigned to the 0-0 transition which is not consistent with a satellite peak on both sides of the main peak. A more comprehensive assignment would be useful.

- I don't follow the authors reason why they cannot use the orbitals from the TDDFT simulations. The decay of the gaussian basis set is different from the decay of the orbital. Gaussian basis function can easily describe the orbitals it just needs more basis function. However, this is the case for modern Gaussian basis sets, e.g., even a STO-3G basis set described the decay of the hydrogen atom wave function correctly. Instead the decay of the orbitals are more determined by the XC functional. Therefore, I would consider changing to the LDA orbitals from Octopus to be a worse description. The decay can be assessed by consider the energy of the HOMO and compare to the IP of the molecule. While Koopman's theory do not hold the correction between HOMO energy and IP do hold for the exact functional. The authors should demonstrate the the orbitals between the two different methods are the same. This is important as they use simulations from three different programs with different functionals.

- I don't understand how the orbitals from Octopus is used in the calculations. In section C, the WPP calculations uses Abinit as far as I can see, although the functional has not been specified. The authors should explain in more detail how the orbitals are used as it is hard to follow as written.

- If I understand the modeling of the charge-transfer dynamics the influence of the tip is modeled as a plane. However, I would expect that the curvature of the tip and more importantly the two tip-atoms needed to bind the molecule are essential for understanding the orbital overlap. Please provide a rationale for why these effects can be ignored.

- For the calculations of the decay rate a single small "atomic" protrusion on the tip is used. This is likely Ok. However, for the molecule to bind to the surface previous work have shown that two metal atoms are necessary and therefore the tip would likely be more complicated. Will this influence the estimates? I understand that the authors are only trying to establish order of magnitude estimates, however, it would be important to understand how sensitive these estimates are to small changes in the calculation setup.

Reviewer #2

(Remarks to the Author)

Review of "Fluorescent single-molecule STM probe" by Friedrich et al. In this study the authors have reported that a PTCDA molecule preserves its intrinsic emission properties even when it is directly attached to a plasmonic scanning probe tip. They analyzed the spectral linewidths and peak positions of current-induced single-molecule emissions and discussed their physical origin. Model calculations were performed to discuss why molecules directly adsorbed on the metal do not lose their luminescence. Based primarily on experimental results, the authors claim that probes with adsorbed luminescent molecules can be used as probes to measure local physical properties.

The techniques used here have been developed in preceding works. The measurements are of high quality and the result of measuring the luminescence of a single molecule directly adsorbed on a metal probe is new and interesting from a basic science viewpoint. On the other hand, unfortunately, no convincing evidence has been presented for the paper's main claim that molecules adsorbed on a probe can be used as a probe (see below for more detail). Theoretical analysis has been done to explain why molecules adsorbed on the probe show luminescence, and there has been no in-depth discussion regarding what this probe can measure.

My assessment is that this study examined the luminescence of molecules adsorbed on the probe. If this research is further extended, there is potential for new probes to be developed, but this has not been demonstrated in this study. Therefore, I do not recommend publication of the manuscript, unless the authors significantly revise the paper or present data to demonstrate the idea of a new probe. Some detailed discussions and criticisms are provided below for improving the manuscript.

1. The authors should discuss in detail what can be measured with this probe. When it comes to the peak energy in luminescence, various factors like the dielectric environment, Lamb shift, and Stark effects play a role. Each of these elements exerts its influence, and without their disentanglement, any attempts at quantitative measurement remain elusive. For example, a peak energy shift of 21 meV was observed on Ag(111) and on NaCl films, but one cannot conclude the origin of this to the dielectric environment without a detailed discussion. The impact of the Lamb shift and the Stark effect with respect to this 21 meV peak shift must be clarified quantitatively.

2. Related to the previous comment, if the peak shift reflects the dielectric environment as the authors claim, one would expect it to vary with position on the NaCl film. It is significant to show that data to demonstrate the usage as a new kind of probe.

3. To what extent is the direction of molecular adsorption determined? How stable is the adsorption structure? This is an important point to measure something quantitatively.

4. If the authors claim that the molecule adsorbed on the probe is an STM probe, as the title suggests, then it is highly desirable to show STM images acquired with that probe.

5. Please specify whether the data in Figure 2d was measured on Ag(111) or NaCl. How do these behaviors differ on Ag and on NaCl?

6. As for the line width argument, I must say that it is also inaccurate. In Ref. 18, it was clearly indicated that EL cannot accurately measure emission linewidths. Ignoring this and describing as if EL can accurately measure line widths is not fair. The effect of the superposition of multiple peaks was discussed in Ref. 18 as an effect of widening the apparent linewidth of an EL peak. A discussion regarding this effect should be added.

7. Why is the luminescence 10 times stronger on NaCl, and how do the intensity, emission peaks, and line widths differ when compared between 2ML and 3ML?

8. It is described that the emission of the PTCDA probe is more intense than the emission of plasmon, but this is not clear because the emission spectrum of plasmon is set to 1/20.

9. Traces during the approach should also be shown in Fig. 1a. What is the applied voltage for Fig. 1a?

Reviewer #3

(Remarks to the Author)

Reviewer #4

(Remarks to the Author)

The manuscript entitled "Fluorescent single-molecule STM probe" by N. Friedrich et al. reports on the functionalization with a 3,4,9,10-perylenetetracarboxylic-dianhydride (PTCDA) molecule of the tip of a low temperature scanning tunneling microscope (STM) acting as a fluorescent probe sensitive to the local electromagnetic and electrostatic fields of samples. The authors demonstrate in a series of scanning tunneling microscopy luminescence (STML) experiments supported by quantum chemical and quasi-electrostatic calculations the variation of the fluorescence of the molecular tip as a function of the targeted substrates. They compare STML spectra of the lifted PTCDA facing the pristine Ag(111) surface and NaCl/Ag(111) with the flat lying PTCDA molecule adsorbed on NaCl/Ag(111). Interestingly, the full width at half maximum (FWHM) of the emission peak is larger using the PTCDA tip in comparison to the flat-lying PTCDA molecule on NaCl using a Ag tip. This demonstrates that the molecule fluorescence is preserved when on the tip due to a reduced charge-transfer rate between the molecule and the tip. They also relate the FWHM to the molecular geometry inducing a dipole close to collinear with the plasmonic electric field in the gap. Last, the emission spectra on top of 2 ML NaCl is also red-shifted by as compared to the bare Ag(111), indicating that the PTCDA tip emission is sensitive to its local dielectric environment. Last, the authors explain using numerical calculations and models the absence of quenching of the suspended molecule's emission. To my opinion, the manuscript is well-written. The experimental data are of very high-quality and supported by advanced numerical calculations. The conclusion of the work is supported by both experimental and numerical data. It is a nice piece of work.

Overall, these experiments follow the general trend in low-temperature scanning probe microscopy, which focus on terminating the apex of an atomic force microscope or an STM with different molecules (CO, PTCDA, nickelocene etc) to access new insights into the structural, electrostatic or magnetic properties of samples. By demonstrating the high spectral sensitivity of such fluorescent probe to its local environment, the authors shed new lights into
Thus, I think this work will attract a broad attention not only for the technical aspects related to the STM/AFM techniques (single-molecule manipulation, STML spectroscopy of single-molecule tips) but also, and maybe more importantly, the physics potentially accessible with such techniques. Therefore, I recommend the publication of the present manuscript in Nature Communication after minor revisions.

Here is my only comment :

1- "Our data also demonstrate that the fluorescent properties of the molecular probe are sensitive to their electromagnetic and electrostatic environment. [...] Further work will be devoted to characterizing the spatial resolution that is achievable with this atomic-scale sensor that could be used, in the close future, in resonant energy transfer microscopy experiments having ultimate lateral and axial precision."

My feeling is that the authors should discuss the term "probe" to avoid confusions. The luminescence of the PTCDA probe is unambiguously sensitive to its local electrostatic environment as demonstrated in the manuscript by varying the sample, the bias voltage or the tip-sample distance. However, combining STM and fluorescent probe such as in the title suggest the combination of both the atomic scale resolution of the STM technique with the high spectral sensitivity of the PTCDA fluorescent tip. My only concerns is that the spatial resolution of the PTCDA tip along the X and Y directions is not demonstrated in the present work. For instance, it would be interesting that the authors provide a cross-section of STML spectra with the PTCDA tip across an NaCl island.

Similar to scanning quantum dot microscopy (SQDM) (Reference 3), the sensitivity of the PTCDA fluorescent probe is related to the electric surface potentials, thus it might be subject to the same lateral resolution. Could the author discuss that? Do they expect submolecular resolution when performing STML maps on molecules with PTCDA tips such as Reference 25? I believe a small discussion along this line would be valuable for the manuscript.

Version 1:

Reviewer comments:

Reviewer #1

(Remarks to the Author)

The authors have addressed all questions raised. The manuscript is recommended for publication.

Reviewer #2

(Remarks to the Author)

2nd Review of "Fluorescent single-molecule STM probe" by Friedrich et al.

I appreciate that the authors considered our comments. The additional experiment around an atomic point defect on the NaCl surface (Fig. 3) is very interesting and seem containing important information. However, our assessment of this study has not been changed much. It examined the luminescence of molecules adsorbed on the probe. Demonstrations as new probes are primitive and are not well explained. Therefore, we cannot recommend publication of this manuscript in the current form. This is mainly because the fundamental question, "What can be measured with this probe?", has not been answered yet. We strongly recommend the authors to reconsider the following points for improving the manuscript.

1. It is obvious, without need for discussion, that molecules change their properties depending on the environment or fields. Therefore, while the fact that a molecule adsorbed on the metal tip shows luminescence is in itself a surprise, it is expected that the resonance properties of the molecule-on-tip change depending on the environment. That's the reason of our assessment that this study examined the luminescence of molecules adsorbed on the probe. If the authors claim that the molecular tip can be used as a new probe, it is very necessary to demonstrate to measure something. Simply showing the property change is not measuring a quantity and is not enough for the claim.

2. The authors discussed the physical origin of 21 meV shift in peak energy in the reply and in the revised manuscript. In the first manuscript it was ascribed to difference in local dielectric environment. Now after revision, it is explained by electrostatic environment (Stark effect). In our opinion, dielectric environment and electrostatic environment are very different. The material surrounding the molecule respond dielectrically to the dynamic movement of charge within the molecule in the excited state. If the material surrounding the molecule changes, its response will change, and thus the molecular resonance frequency is expected to be modulated. This is the effect of the dielectric environment, which is expected to result in a shift in peak energy when the molecular tip is on Ag and NaCl.

Electrostatic field effect (Stark effect) is a shift in resonance energy due to an electrostatic field.

The authors have changed their explanation, suggesting that not enough thought has been given to the key result, the energy shift. Now in the revised manuscript, the dielectric environment effect is completely ignored. As I mentioned in the previous comment, there are THREE, not two, possible causes for the peak energy shift; the dielectric environment, Lamb shift, and Stark effects. It is necessary to discuss the impact of all effects on the 21 meV energy shift, rather than ignoring them without rational reason. We would not recommend publication unless the physical origins of the energy shift are adequately discussed.

3. The additional experiment around an atomic point defect on the NaCl surface (Fig. 3) is very interesting. However, the results are not discussed at all, unfortunately. If the authors cannot discuss the results because the defects are unknown, then the results are not worthy of publication. At the very least, they should measure something known. Could that defect be a Cl vacancy in NaCl? Cl vacancies have been investigated in previous STM studies and can be created on NaCl films with an STM tip, so I think they could be considered a sample with well-understood properties. Since these results are an important demonstration, they should be revised and included in the paper.

4. The data in Figure 3 are very interesting. The red shift is larger at a little distance from the molecule than at the closest point to the molecule. Why is this? Also, what is the orientation of PTCDA in this case? Since the molecule is anisotropic, the orientation of the molecule must be known in order to interpret these results.

5. Was the STM image in Fig. 3b measured with the molecular tip or metal tip?

6. In response to my previous comment "If the authors claim that the molecule adsorbed on the probe is an STM probe, as the title suggests, then it is highly desirable to show STM images acquired with that probe.", they responded "We yet did not succeed in acquiring a full STM image together with spectral acquisition with a PTCDA-tip."

I am not asking for something complicated. I am simply saying that STM images obtained with a PTCDA-tip should be shown. If the author refers to a PTCDA-tip as an STM tip, it is required.

Reviewer #3

(Remarks to the Author)

Reviewer #4

(Remarks to the Author)

The authors have addressed the comments and corrections satisfactorily.

Version 2:

Reviewer comments:

Reviewer #2

(Remarks to the Author)

3rd Review of "Fluorescent single-molecule STM probe" by Friedrich et al.

We are disappointed in the authors' decision not to discuss one of the most important results of the paper, the 21 meV energy shift between on Ag and on NaCl, just by saying it is beyond the scope of the paper. I believe that discussing the physical origins of the energy shift is not beyond the scope of the paper, but rather should be the core of the paper. Because without that discussion, one cannot conclude what this probe can measure.

We cannot recommend publication of this manuscript in the current form. This is mainly because the fundamental question, "What can be measured with this probe?", has not been answered yet. We strongly recommend the authors to reconsider the following points for improving the manuscript.

1. Since the authors do not seem to be aware of the well-known phenomenon of dielectric-induced emission peak shifts, I list some examples of papers dealing with the effect.

[1] F. J. R. Costa, T. G.-L. Brito, I. D. Barcelos, and L. F. Zagonel, Impacts of Dielectric Screening on the Luminescence of Monolayer WSe₂, *Nanotechnology* 34, 385703 (2023).

[2] S. Borghardt, J.-S. Tu, F. Winkler, J. Schubert, W. Zander, K. Leosson, and B. E. Kardynał, Engineering of Optical and Electronic Band Gaps in Transition Metal Dichalcogenide Monolayers through External Dielectric Screening, *Phys. Rev. Mater.* 1, 054001 (2017).

[3] M. Florian, M. Hartmann, A. Steinhoff, J. Klein, A. W. Holleitner, J. J. Finley, T. O. Wehling, M. Kaniber, and C. Gies, The Dielectric Impact of Layer Distances on Exciton and Trion Binding Energies in van Der Waals Heterostructures, *Nano Lett.* 18, 2725 (2018).

It is well known that the presence of a dielectric material in close proximity to the emitting material can shift the emission peak by several 10 meV. I believe this effect cannot be ignored in the explanation of the 21 meV energy shift, since the materials surrounding the molecule are different on Ag or on NaCl.

Now in the manuscript, the dielectric screening effect from the surrounding is completely ignored. As I mention continuously in the comment, there are three, not two, possible causes for the peak energy shift; the dielectric environment, Lamb shift, and Stark effects. It is necessary to discuss the impact of all effects on the 21 meV energy shift, rather than ignoring them without rational reason. We would not recommend publication unless the physical origins of the energy shift are adequately discussed.

2. Since the authors seem unwilling to explain the newly added data (Fig. 3), we strongly recommend that the data be excluded from the paper. If the data is measured without knowing what kind of substance or physical quantity is being measured, what is the meaning of the data?

I am not making the comment for my own satisfaction. I am giving advice to improve the paper.

Here is my comment in the first round, "If the peak shift reflects the dielectric environment as the authors claim, one would expect it to vary with position on the NaCl film. It is significant to show that data to demonstrate the usage as a new kind of probe."

This comment points out that it is important to show nanoscale probe position dependence for claiming usage as a STM probe. Of course, the comment assumed that if the results are to be shown, the meaning should be explained.

Reviewer #3

(Remarks to the Author)

Version 3:

Reviewer comments:

Reviewer #2

(Remarks to the Author)

4th Review of "Fluorescent single-molecule STM probe" by Friedrich et al.

The authors have newly discussed the screening effect of the NaCl film in the optical frequency domain, and I appreciate them for improving the quality of the paper by providing a more detailed description of the observed phenomena. This fluorescent probe is inherently sensitive to both the electrostatic field and the electric field in the optical frequency domain. In other words, the energy shift of the emission peak is caused by both the electrostatic field and the optical frequency electric field. Compared to the previous manuscript, which mainly discussed only the Stark effect, the current version describes the physical phenomena more accurately. I believe this description is crucial for understanding the characteristics of the new probe.

There have already been studies using STM probes with PTCDA molecules adsorbed at the tip to probe electrostatic fields, electrostatic potentials, and magnetic fields.

[1] Wagner, C., Green, M. F. B., Leinen, P., Deilmann, T., Krüger, P., Rohlfing, M., Temirov, R., & Tautz, F. S. (2015).

Scanning Quantum Dot Microscopy. *Physical Review Letters*, 115(2), 026101.

<https://doi.org/10.1103/PhysRevLett.115.026101>

[2] Wagner, C., Green, Matthew. F. B., Maiworm, M., Leinen, P., Esat, T., Ferri, N., Friedrich, N., Findeisen, R., Tkatchenko, A., Temirov, R., & Tautz, F. S. (2019). Quantitative imaging of electric surface potentials with single-atom sensitivity. *Nature Materials*, 18(8), 853–859. <https://doi.org/10.1038/s41563-019-0382-8>

[3] Esat, T., Borodin, D., Oh, J., Heinrich, A. J., Tautz, F. S., Bae, Y., & Temirov, R. (2024). A quantum sensor for atomic-scale electric and magnetic fields. *Nature Nanotechnology*. <https://doi.org/10.1038/s41565-024-01724-z>

[4] Bolat, R., Guevara, J. M., Leinen, P., Knol, M., Arefi, H. H., Maiworm, M., Findeisen, R., Temirov, R., Hofmann, O. T., Maurer, R. J., Tautz, F. S., & Wagner, C. (2024). Electrostatic potentials of atomic nanostructures at metal surfaces quantified by scanning quantum dot microscopy. *Nature Communications*, 15(1), 2259. <https://doi.org/10.1038/s41467-024-46423-4>

If the authors claim, as in the current manuscript, that their fluorescent probe is primarily sensitive to the electrostatic field, it is highly desired to compare its performance and versatility with those of previous studies. This comparison would help readers appreciate the significance of developing this probe. Conversely, if the probe does not demonstrate superiority over prior research, it would not merit publication in *Nature Communications*.

On the other hand, the ability to probe electric fields in the optical frequency domain has not been reported, and I believe this aspect is highly valuable. I recommend adding a discussion and perspective on the development of a qualitatively different new probe.

As for Fig. 3, still I believe it should be excluded from the paper.

I am not asking for a full explanation of everything. However, at the very least, I believe it is necessary to provide information on what physical quantities are being measured or what substances are being examined. At present, both of these pieces of information are insufficient.

In the aforementioned prior studies [1-4] using similar PTCDA probes, the substances being measured are confirmed at the atomic level. The physical quantities being measured are also discussed. Therefore, what I am requesting is not unreasonable, but rather a standard level of detail expected when claiming the development of a new probe.

In summary, if the authors continue to assert that the probe is only sensitive to the electrostatic field, they must demonstrate its advantages over previous studies. Therefore, in its current form, I cannot recommend the manuscript for publication.

Reviewer #3

(Remarks to the Author)

I co-reviewed this manuscript with one of the reviewers who provided the listed reports. This is part of the *Nature Communications* initiative to facilitate training in peer review and to provide appropriate recognition for Early Career Researchers who co-review manuscripts.

REVIEWER COMMENTS

Reviewer 1: In this work the authors demonstrate fluorescence from a single molecule bound to a STM tip. Normally, the fluorescence of molecules on metal surfaces are quenched as the molecule tends to lie flat on the surface and thus have significant interactions with the surface. Here, the authors avoids this by using a PTCDA molecule which has previously been demonstrated to adopt a standing-up configuration on the tip due to electrostatic forces. They further demonstrate that the FWHM and fluorescence energy is sensitive to its local environment and thus could be used as a probe of the local electrostatic environment. A theoretical model explaining the results, in particular, why the fluorescence is not quenched is presented. Overall, this is a high quality work that demonstrates an important advance in the field and thus is well suited for Nature Communications. Below I have some technical questions that the authors should address before publication.

We thank the reviewer for their feedback and appreciate that they consider our work to be of "high quality" and "an important advance in the field".

- It is assumed that the PTCDA molecule remain standing up with the transition dipole movement aligned with the near-field in the junction. While this is reasonable it would be important to characterize/estimate the influence of this assumption. Depending on the local environment one could expect that the molecule binds with different angles if the binding strength is not too large. This would be particularly important if this will be used as a probe of the local environment at envision by the authors. Could the authors provide an estimate of the binding strength and the influence of the alignment on the quenching?

We thank the reviewer for this suggestion. Indeed, it was reported that the molecule is held in the upright configuration by weak dispersive forces and the molecule is therefore prone to tilting. We have therefore considered the effect of a tilting angle on the charge transfer rate γ_{ch} of LUMO and the plasmon-induced decay rate γ_{pl} . We show these results in Fig. R1. We considered two tilting geometries. First we considered that the molecule pivots around an axis perpendicular to the molecular plane that passes through one of the lateral oxygen atoms (Fig. R1b,d). In the second configuration the molecule was rotated around the axis passing through the two lateral oxygen atoms (Fig. R1c,e). The effect of tilting leads to relatively minor modifications of the charge transfer rate γ_{ch} for the moderate angles (up to $\sim 45^\circ$) used in the calculation, although a steady increase of γ_{ch} is observed for the geometry

FIG. R1: Plasmon-induced decay rate γ_{pl} and charge-transfer rate γ_{ch} of LUMO calculated as a function of the molecule tilt angle for the geometries shown in (b)-(e).

shown in Fig. R1e (black dots). We note that the data for zero tilt angle correspond to the decay rate calculated for the shortest tip-molecule distance shown in Fig. 4b of the main text. For γ_{pl} we observe a steady cosine-like decrease for both geometries. We note that for the tilt angles considered here γ_{pl} prevails over γ_{ch} and quenching of trion light emission due to charge-transfer processes is therefore not expected.

Thus, we believe that our qualitative conclusions on the respective role of the electronic and electromagnetic channels in the trion decay, as follows from the results shown in Fig. 4 of the main text, are robust with respect to the possible molecular tilt.

Action: We included the corresponding discussion and figure in Section S2.G of the SI. While performing the new calculations we have also noticed that the original data in Fig. 4e of the main text were not fully converged with respect to the mesh size. We have therefore updated the data which now show a plasmon-induced decay rate of up to 24 meV compared to the original value of up to 21 meV.

- In Figure 2a the authors shows a shifts of 21meV for the emission as the tip is brought closer to the NaCl substrate. However, the energy is lower that that of the molecule lying flat on the NaCl substrate and thus should have a larger perturbation. Why is this?

The reviewer is right in pointing out that the molecule lying flat on the NaCl substrate has a higher exciton emission energy than the lifted molecule. However, it is important to point out that the molecule adsorbed on the substrate is affected by a number of effects that are different from the ones for the molecule on the tip. The flat-adsorbed molecule has slightly altered geometry as it is bent due to the interaction with the substrate. It also experiences different screening due to the proximity of NaCl and the underlying metal surface. This screening can be both static (image-charge effects are expected due to the negative charge of the molecule) and dynamical (the plasmonic Lamb shift). The flat-lying molecule is also differently exposed to the electrostatic field induced by the applied bias in the gap: (1) the orientation of the molecule with respect to the field lines, and (2) the gap size differs for a similar current setting. For all the reasons above, it is difficult to properly evaluate a relatively small spectral shift of the exciton spectral line between the two configurations.

- The two satellite peaks (ν_1) is assigned as coupling to molecular vibrations. The authors could easily assign this by calculating the vibronic effect using the TDDFT module. In the simulations the main peak is assigned to the 0-0 transition which is not consistent with a satellite peak on both sides of the main peak. A more comprehensive assignment would be useful.

The reviewer is right that a vibronic progression can be observed on both sides of the zero-phonon line (ZPL). Similar spectra have been observed and discussed for the same molecule by other authors [Paulheim et al., Phys. Chem. Chem. Phys. (18), 32891-32902 (2016), Kimura et al., Nature (570), 210-213 (2019), Doležal et al., ACS Nano 16(1), 1082-1088 (2022)]. We attribute the "anti-Stokes" peak to the hot emission emerging from a vibrationally excited molecule. To further support the claim that vibronic peaks are observed, we calculated a vibronic spectrum of PTCDA using the FCVT module implemented in Gaussian 16 rev. C.01 and using the B3LYP/AUG-cc-pVDZ level of theory at T=0K and including an artificial broadening of 10 cm⁻¹. In Fig. R2 we compare the calculated spectrum (black line) with an experimentally obtained spectrum (red line) of a PTCDA molecule adsorbed flat on NaCl. The calculated spectrum shows that the ZPL is dominating the spectrum as the vibronic activity in PTCDA is relatively low. This finding confirms our assignment of the most intense peak as the ZPL of the molecular emission. In the inset we also show

the calculated vibrational modes that correspond to the most intense peaks of the vibronic shoulder that can be straightforwardly assigned to the experimental peaks. The vibronic peaks appearing at -231 cm^{-1} and $+238\text{ cm}^{-1}$ are those observed in the spectra of the lifted PTCDA, and can be assigned to the in-plane breathing mode of the molecule. The peaks of lower energy (56 cm^{-1} and 129 cm^{-1}) are not present in the calculated spectrum and can be attributed to vibrational modes that arise from the interaction of the molecule with the substrate, as discussed in [Paulheim et al., Phys. Chem. Chem. Phys., 18, 32891-32902 (2016)].

The exact origin of the vibrational pumping mechanism leading to hot luminescence may originate from inelastic electron tunneling as well as vibrational pumping related to the sequential charging of the molecule and subsequent photon emission through e.g. the Franck-Condon mechanism. Unraveling the exact details of this process is beyond the scope of our manuscript.

Action: We included the corresponding discussion and figure in Section S2.F of the SI.

- I don't follow the authors reason why they cannot use the orbitals from the TDDFT simulations. The decay of the gaussian basis set is different from the decay of the orbital. Gaussian basis function can easily describe the orbitals it just needs more basis function. However, this is the case for modern Gaussian basis sets, e.g., even a STO-3G basis set described the decay of the hydrogen atom wave function correctly. Instead the decay of the orbitals are more determined by the XC functional. Therefore, I would consider changing to the LDA orbitals from Octopus to be a worse description. The decay can be assessed by consider the energy of the HOMO and compare to the IP of the molecule. While Koopman's theory do not hold the correction between HOMO energy and IP do hold for the exact functional. The authors should demonstrate the the orbitals between the two different methods are the same. This is important as they use simulations from three different programs with different functionals.

As for the use of orbitals from Octopus, we apologise to the reviewer for the confusion caused by our mistake. Calculations using Octopus have not been used in the final submitted version of the manuscript and the reference to Octopus was mistakenly forgotten in the text.

Nevertheless, we thank the reviewer for raising the question of usability of orbitals represented by Gaussian basis sets (or finite localized basis sets in general). Orbitals calcu-

FIG. R2: (a) Photon emission spectrum obtained experimentally (red line) and theoretically (black line) calculated for zero temperature. The most intense vibronic peaks are labelled with their resonance energy. The black labels correspond to the vibrational frequencies calculated for a molecule in vacuum using Gaussian 16, the red labels correspond to the experimental positions of the peaks. The energy scale is defined relative to the ZPL energy. Acquisition parameters for the experimental spectrum: $V = -2.5$ V, $I = 60$ pA, $t = 30$ min. (b) Calculations shown in [Paulheim et al., Phys. Chem. Chem. Phys., 18, 32891-32902 (2016)] that can be assigned to the low-energy vibrational peaks that arise from substrate-molecule interactions.

lated as linear combinations of such basis functions cannot exactly fulfill the Schrödinger's equation in the vacuum region far from the atoms of the molecule as their mathematical form does not exactly reflect the orbital energy (or more precisely, the ionization energy or electron affinity of the molecule). They can reasonably approximate the spatial extent of the orbitals in regions close to the molecule (and are therefore good basis functions for the variational method - DFT), but their asymptotic behaviour necessarily obeys the Gaussian decay compared to the much slower decay expected from the exact solution.

Action: We have removed the reference to calculations of molecular orbitals using Octopus. We have also added in the SI the explicit description of the exchange-correlation functional used in the Abinit package, and we added the corresponding reference to the

Perdew-Wang work.

- I don't understand how the orbitals from Octopus is used in the calculations. In Section S2.C, the WPP calculations uses Abinit as far as I can see, although the functional has not been specified. The authors should explain in more detail how the orbitals are used as it is hard to follow as written.

We apologise for the confusion. In the submitted version of the supplementary information we still referred to the orbitals calculated using Octopus, but these calculations have not been included in the final version of the manuscript nor supplementary material as we use the more accurate model based on wave-packet propagation.

The local density approximation with Perdew-Wang correlation energy was used for Abinit calculations as we now explicitly state in SI.

Action: We have removed the reference to calculations of molecular orbitals using Octopus. We have also added in the SI the explicit description of the exchange-correlation functional used in the Abinit package, and we added the corresponding reference to the Perdew-Wang work.

- If I understand the modeling of the charge-transfer dynamics the influence of the tip is modeled as a plane. However, I would expect that the curvature of the tip and more importantly the two tip-atoms needed to bind the molecule are essential for understanding the orbital overlap. Please provide a rational for why these effects can be ignored.

We thank the reviewer for this comment. As the reviewer points out, in previous works (Refs. [29,30]) it was reported that the PTCDA molecule standing upright on a perfect flat Ag surface is attached to a pair of silver atoms. In our experiment, however, the molecule is attached to a tip of an unknown geometry and atomic structure, it is therefore unclear what the exact adsorption geometry is. The tip curvature would likely have a small effect on the charge-transfer rate as the charge is injected from the molecule into the metal substrate almost perpendicularly to the surface. We show this effect in Fig. R3, where we calculate the resonance wave function of the LUMO, LUMO+1, and LUMO+2 orbitals coupled to the surface in both the perpendicular configuration (Fig. R3a-c) and the parallel configuration (Fig. R3d-f). As we show, in the perpendicular configuration most of the electron flow could be contained in a cylinder with an axis aligned with the molecule and a radius of only

$\approx 1\text{nm}$. We therefore anticipate that the molecule would not be very sensitive to large-scale geometry effects such as the tip curvature. If present, the effect of the (positive) tip curvature can only lead to a lower charge transfer rate as, overall, the metal will appear more distant from the molecule. On the other hand, the situation where the molecule is attached to a pair of silver atoms, which are in turn adsorbed on silver surface (we replaced two atoms by a complete surface plane thus approaching the metal bulk to the molecule) would likely lead to an overall decrease in the orbital overlap between the molecule and the surface. We therefore think that by choosing the most simple geometry where the molecule is directly interacting with a generic planar interface we maximize the charge transfer rate, which enhances quenching of the molecular exciton. Our finding that even in this case the photon emission is possible, appears then as a robust result.

Action: We extended the discussion of the wave packet propagation approach in the SI (section S2.D) to explain the calculation of the resonance wave functions. In the SI we have also included the corresponding figure and the discussion of the resonant electron transfer into the metal for different geometries and molecular orbitals.

- For the calculations of the decay rate a single small "atomic" protrusion on the tip is used. This is likely Ok. However, for the molecule to bind to the surface previous work have shown that two metal atoms are necessary and therefore the tip would likely be more complicated. Will this influenced the estimates? I understand that the authors are only trying to establish order of magnitude estimated, however, it would be important to understand how sensitive these estimates are to small changes in the calculation setup.

Following the suggestion of the reviewer we have performed further calculations using a different kind of protrusion reflecting the possibility of having e.g. two atoms forming the protrusion. Concretely, we have used a single protrusion of a radius varying between $r_p = 0.2\text{ nm}$ and 0.5 nm , and a double protrusion composed of a pair of overlapping hemispherical protrusions as shown in Fig. R4a,b. The molecule was placed in the middle of the gap in the vertical orientation and the gap was held constant for all protrusion geometries. Owing to that, the resonance frequency of the plasmon varies slightly when different protrusions are considered (Fig. R4c,d). For this reason, we pick the maximum of the calculated plasmon-enhanced broadening for each protrusion geometry and compare these maxima in Fig. R4e. We note that the slight frequency shift could in principle be compensated by adjusting the

FIG. R3: Electron probability distribution at orbital resonances calculated for a PTCDA molecule (a-c) standing upright on a jellium Ag surface, and (d-f) lying flat on the metal substrate, shown for (a,d) LUMO, (b,e) LUMO+1 and (c,f) LUMO+2. At the start of the propagation, the wave function has the form of the respective orbital of the unperturbed molecule. The molecule is at a distance of $5 a_0$ from the image plane of the surface marked by the dashed line. The color represents the square modulus of the wave function which is displayed $y = 2.5 a_0$ above the $y = 0$ plane, which is chosen to be a symmetry plane of the molecule. The scale bar is 1 nm. In all cases the dynamical range of the color scale is e^{-12} to 1.

geometry (total length) of the model tip and therefore does not play a significant role. For reference, we also plot the position of the plasmon resonance for each geometry in Fig. R4f. The Purcell effect (plasmon-enhanced broadening) was calculated as in Fig. 4e of the main text. The results calculated for the modified protrusion show only a minor deviation from the data where a single-atom protrusion was considered. It therefore appears that the precise nature of the protrusion does not play a significant role.

Action: We have included the corresponding figure and the discussion in the SI (section S2.E).

FIG. R4: The influence of the picocavity shape on the plasmon-enhanced decay. (a,b) Two picocavity shapes have been taken into account: (a) a single hemispherical protrusion, and (b) a double protrusion. The radius r_p of these protrusions is varied and for each geometry the spectrum of the plasmon-induced trion broadening is calculated as discussed in the Supplementary Information. (c,d) the spectra corresponding to the geometries shown in (a,b). (e) Comparison of the calculated maxima and (f) position of the plasmon resonance of the spectra in (c,d).

Reviewer 2: Review of "Fluorescent single-molecule STM probe" by Friedrich et al. In this study the authors have reported that a PTCDA molecule preserves its intrinsic emission properties even when it is directly attached to a plasmonic scanning probe tip. They analyzed the spectral linewidths and peak positions of current-induced single-molecule emissions and discussed their physical origin. Model calculations were performed to discuss why molecules directly adsorbed on the metal do not lose their luminescence. Based primarily on experimental results, the authors claim that probes with adsorbed luminescent molecules can be used as probes to measure local physical properties. The techniques used here have been developed in proceeding works. The measurements are of high quality and the result of measuring the luminescence of a single molecule directly adsorbed on a metal probe is new and interesting from basic science viewpoint. On the other hand, unfortunately, no convincing evidence has been presented for the paper's main claim that molecules adsorbed on a probe can be used as a probe (see below for more detail). Theoretical analysis has been done to explain why molecules adsorbed on the probe show luminescence, and there has been no in-depth discussion regarding what this probe can measure.

My assessment is that this study examined the luminescence of molecules adsorbed on the probe. If this research is further extended, there is potential for new probes to be developed, but this has not been demonstrated in this study. Therefore, I do not recommend publication of the manuscript, unless the authors significantly revise the paper or present data to demonstrate the idea of a new probe. Some detailed discussions and criticisms are provided below for improving the manuscript.

We thank the reviewer for their detailed reading of the manuscript, and for recognizing the high quality and the novelty of the reported data. More precisely, the reviewer acknowledges that preserving the fluorescence properties of a molecule directly attached to a molecular tip constitutes, in itself, an important progress. They found our work too preliminary, however, when it comes to using our molecular tip as a probe. Here, we disagree with the reviewer's comment as the probing capacity of the PTCDA-tips is demonstrated in several occasions in the original version of the manuscript, something that is acknowledged by the other referees and that we will detail in our response below. However, we fully agree that this aspect needs to be reinforced. In response to the referee's comments we conducted additional experiments (see Figure 3 of the amended manuscript and the corresponding text, see also Figure R5 in the present reply) where the fluorescence

response of the molecular sensor is registered as a function of its distance with respect to an impurity on NaCl. This set of data reveals a red-shift of the 0-0 fluorescence line of the molecule-tip when it is at a nanometer distance from the defect. We believe that this constitutes a more convincing proof-of-principle of the sensor capability of our molecular-tip.

1. The authors should discuss in detail what can be measured with this probe. When it comes to the peak energy in luminescence, various factors like the dielectric environment, Lamb shift, and Stark effects play a role. Each of these elements exerts its influence, and without their disentanglement, any attempts at quantitative measurement remain elusive. For example, a peak energy shift of 21 meV was observed on Ag(111) and on NaCl films, but one cannot conclude the origin of this to the dielectric environment without a detailed discussion. The impact of the Lamb shift and the Stark effect with respect to this 21 meV peak shift must be clarified quantitatively.

The referee's comment is correct when they write that both the Lamb effect and the Stark effect (which encompasses effects related to the dielectric environment) can influence the peak energy of the main line. The 21 meV shift reported on NaCl can, in principle, be associated to any of these two effects. Note first, that independently of its "Lamb" or "Stark" origin, this shift reflects a change in the molecular-tip environment, confirming that it can indeed be used as a molecular probe. In the paper, a detailed discussion of the respective impact of Lamb and Stark effect is provided based on Fig 2d and e. Here, the voltage-dependent shift of the peak at constant tip-molecule height can *only be assigned* to a Stark effect, as the plasmon-exciton coupling (responsible for Lamb shifts) remains constant. More quantitatively, we observe an almost linear red shift of ~ 12 meV when the voltage is increased from 2.5 V to 2.9 V. When the molecule initially lifted above a metal surface is transferred on top of NaCl, its electrostatic environment is influenced by the change of the local work function of the substrate. By covering the Ag(111) surface with NaCl its work function is decreased by ≈ 1 eV [Imai-Imada et al., Phys. Rev. B, 98, 201403 (2018)], which increases the electric field in the tip-substrate gap as a direct consequence, equivalently to adding ≈ 1 V of external bias. By extrapolating the experimental values obtained for the Stark shift we therefore conclude that the exciton line should red shift by ~ 30 meV, which is a value consistent with the observed shift of ~ 21 meV. We therefore believe that the tentative assignment of this 21 meV red shift to the change of the electrostatic environment

is reasonable.

Based on this semi-quantitative observation, and on a previous paper that also concluded on a moderate impact of Lamb shift (Ref. 15), we conclude that the shift of the molecular peak is dominated by the local electrostatic environment. The comment of the reviewer made us realize 1- that it is not sufficiently clear in our manuscript that the Stark effect and the effect of the dielectric environment are associated to the same physical effect, 2- to which extend we can show that our probe is sensitive to electrostatic field and 3- that the qualitative character of our argumentation may not be sufficiently clear.

Action: - With respect to point 1, we decided to modify the sentence (end of page 3): "This shift suggests that the emission properties [...] are dependent on its dielectric environment" by "This shift suggests that the emission properties [...] are dependent on its electrostatic environment (Stark shift)" to avoid any misunderstanding.

- With respect to point 2, we modified the discussion of fig 2e,d in page 4 that now reads: "The voltage-dependent shift of the peak at constant tip-molecule height can be assigned to the sole Stark effect due to the static field E_V in the junction $\mu - \mu_0 = -c_S E_V$ (with c_S being a constant), as the plasmon-exciton coupling (responsible for Lamb shifts) remains constant. Figure 2e reveals that this line-shift evolves essentially linearly with distance z , at a rate of ≈ 50 meV/nm, and with voltage, at a rate of ≈ 30 meV/V. Considering that $E_V = V/z$ [V/nm], these values are consistent with each other yielding $c_S \approx 0.06 e \cdot \text{nm}$. These findings are also close to the rates reported for phthalocyanine molecules lying flat on NaCl and addressed by a metal tip [15]. In this case, the line shifts caused by changes in the tip-sample distance and voltage were also dominated by the Stark effect due to the electrostatic field in the STM junction. The same conclusion can be drawn for the 21 meV shift observed when the PTCDA-tip is moved on top of 2ML- NaCl (Fig. 2a), which can be assigned to changes of the local work function of the substrate. Indeed, covering the Ag(111) surface with NaCl leads to a work function decrease by ≈ 1 eV [41], which is equivalent to adding ≈ 1 V of external bias. By extrapolating the experimental values obtained in Fig. 2e, we conclude that the exciton line should redshift by ≈ 30 meV on NaCl, consistent with the observed ≈ 21 meV shift. To further illustrate the sensitivity of the PTCDA-probe to changes in its local environment, we recorded STML spectra for each pixel of a line-scan for a PTCDA-tip that is laterally moved toward an atomic-scale defect located on a NaCl bilayer (Fig. 3). The spectra reveal a small, but observable, shift (≈ 3 meV) of the spectral

line in the proximity of the defect. Together with the data of Fig. 2a, d, e, this demonstrates that one can use the spectral shift of the PTCDA tip emission line as a probe of the local electrostatic field.”

- With respect to point 3, besides the modifications mentioned just above, we also modified the conclusion that now reads: ”While our data also demonstrate qualitatively that the fluorescent properties of the molecular probe are sensitive to their electromagnetic and electrostatic environment, additional work will be required for a full quantitative description of all the effects.”

2. Related to the previous comment, if the peak shift reflects the dielectric environment as the authors claim, one would expect it to vary with position on the NaCl film. It is significant to show that data to demonstrate the usage as a new kind of probe.

We thank the reviewer for their suggestion. On a flat NaCl layer, and in the absence of defects, we do not expect measurable changes of the dielectric (i.e. electrostatic) environment at a distance of 1 nm from the surface corresponding to the position of our ”probe” (see eqs 7,8 of [Phys. Rev. A 67, 032901 (2003)] (<https://doi.org/10.1103/PhysRevA.67.032901>) for the dependence of the field of the ± 1 point charge lattice on the distance from its surface). This should, however, not be the case in the presence of surface (or sub-surface) defects. Following the reviewers’ suggestions, we recorded STML spectra for each pixel of a line-scan for a PTCDA-tip that is progressively approached to an atomic-scale defect located in a NaCl bilayer (see Fig.R5 and Figure 3 of the amended manuscript and the corresponding text). The spectra reveal a small, but clearly observable, shift of the spectral line in the proximity of the defect. The unknown nature of the defect, as well as the limited number of acquired data points, prevents a full quantitative analysis of the spectral shift, but this new set of data perfectly illustrates the potential of the PTCDA-tip that is sensitive to atomic-scale variation of its environment.

Action: We have added a new figure to the main text (Fig. 3) and related descriptions and discussions (primarily in page 4).

3. To what extent is the direction of molecular adsorption determined? How stable is the adsorption structure? This is an important point to measure something quantitatively.

We agree with the reviewer that the adsorption configuration of the molecule on the tip

FIG. R5: **a** Constant height STML spectra ($V = 2.3\text{V}$, $t = 1\text{ min/spectrum}$) of a PTCDA tip acquired at different lateral distances (see dots in (b)) from a defect on 2ML NaCl. (b) STM image ($I = 5\text{ pA}$, $V = -0.5\text{ V}$) of the defect.

may affect its spectral response. However, for a given adsorption configuration (i.e. contact to a single atom or two atoms of the tip), the spectral response is expected to be stable, and so even if the *absolute* width and energy position of the peaks may be affected by the tip adsorption configuration, their *changes* in widths and energy shifts as a function of tip position reflect the tip environment. Anyway, we theoretically tested several new configurations, including different directions/orientations of the molecule and tip/molecule bondings, Fig.R1 and Fig.R4, allowing one to see, for example, the effect of a reduced collinearity between plasmon electric field and the dipole of the exciton on peak width. More details can be found on that point in our respond to Reviewer 1.

Additional discussion and calculations have been added in SI (Section S2.E and S2.G) showing the effect of different molecular adsorption configurations on the tip.

4. If the authors claim that the molecule adsorbed on the probe is an STM probe, as the title suggests, then it is highly desirable to show STM images acquired with that probe.

We yet did not succeed in acquiring a full STM image together with spectral acquisition with a PTCDA-tip. These images require very long acquisition times, and extreme tip stability. Instead, we propose a line-scan acquired in the proximity of a defect on the NaCl layer (see Figure 3 of the amended manuscript and the corresponding text on page 4, see also Figure R5 in the present reply). We hope that this set is sufficiently convincing to

validate the proof-of-principle of the molecular probe.

Action: We have added a new figure to the main text (Fig. 3) and related descriptions and discussions (primarily in page 4).

5. Please specify whether the data in Figure 2d was measured on Ag(111) or NaCl. How do these behaviors differ on Ag and on NaCl?

These data sets were obtained on Ag(111). We thank the reviewer for noticing that this information was absent in the caption of the figure. We corrected this issue in the amended manuscript by adding the corresponding description in the caption of Fig. 2. For reviewer's convenience we report in Fig. R6 the voltage and distance dependencies of the STML spectra for a PTCDA-tip in front of NaCl. The spectral shifts are similar to those reported in Fig. 2 and e for the PTCDA-tip located on top of Ag(111).

Action: the caption of figure 2 now reads: "d X^- emission spectra for varying PTCDA-tip to Ag(111) separation." Moreover, Fig. R6 has been added in section S1.C.

6. As for the line width argument, I must say that it is also inaccurate. In Ref. 18, it was clearly indicated that EL cannot accurately measure emission linewidths. Ignoring this and describing as if EL can accurately measure line widths is not fair. The effect of the superposition of multiple peaks was discussed in Ref. 18 as an effect of widening the apparent linewidth of an EL peak. A discussion regarding this effect should be added.

It is true that many effects can contribute to the width of an emission line. We discuss in detail several of those contributions on page 3, but indeed did not mention the possible presence of multiple peaks as discussed in Ref. 18. To be fully complete, we now explicitly acknowledge this specific contribution in the amended manuscript. However, this contribution of approx 4 meV reported in Ref. 18 is surprisingly higher than that found in other publications (see Ref. 15). Moreover, it is (i) smaller than the dephasing contribution to the linewidth (leading to the 7 meV limit reported in Fig 2d), and is fully negligible (ii) at or close-to resonance conditions between plasmons and trion, *i.e.*, *the precise experimental conditions at which we claim PTCDA-tips can be used as a probe of the local electromagnetic field*. Note, eventually, that the measured effect of the plasmon-exciton coupling on the FWHM of the PTCDA emission (flat or on the tip) is *quantitatively* reproduced by our theory. In fact, we believe that the data of Fig. 2b and c constitute a strong argument to

FIG. R6: **a** X^- emission spectra for varying tip-substrate separation (left panel, $V = 2.5$ V) and for varying bias voltage (right panel, $z = 2.7$ nm) for a PTCDA-tip on top of 2ML-NaCl. A red shift of the central emission is observed upon reducing z or increasing V . **b** Relative shift of the central emission line μ extracted from the data in **a**, taking the reference energy μ_0 from the bottom spectrum in **a**. Spectra in **a** are offset vertically for easier comparison. These results are similar to those observed for the PTCDA-tip on top of Ag(111) in Fig. 2d and e.

claim that PTCDA attached to the tips can be used as a probe of their electromagnetic environment.

Action: the possible presence of multiple peaks in our PTCDA-tip spectra (as discussed in Ref. 18) is now explicitly mentioned in page 3 in the amended manuscript. We have also modified the discussion of the plasmon-trion strong coupling condition on page 4 of the manuscript, it now reads: "Assuming that the trion couples to a single plasmon mode as described in Ref. [15], we estimate g for the resonant case (top spectrum Fig. 2b) from the plasmon-induced width $\hbar\gamma_{\text{pl}} \approx 46$ meV (assuming that ≈ 7 meV of the total ≈ 53 meV broadening accounts for other mechanisms) and $\hbar\kappa \approx 100$ meV (\hbar is the reduced Planck constant) as $\hbar g = \hbar\sqrt{\gamma_{\text{pl}}\kappa}/2 \approx 34$ meV. Since $2g$ is smaller than κ we conclude that the plasmonic losses still dominate over the plasmon-trion coupling and we therefore observe a

weak trion-plasmon coupling.”

7. Why is the luminescence 10 times stronger on NaCl, and how do the intensity, emission peaks, and line widths differ when compared between 2ML and 3ML?

The stronger emission on NaCl is a reproducible observation, and may indeed seem surprising. While a full explanation of this effect is currently missing, we tentatively ascribe it to the different excitation path on top of NaCl and on top Ag(111) (see Supp mat Section S2.H). Unfortunately, the PTCDA-tip was extremely unstable when we attempted to measure on 3ML NaCl, preventing the acquisition of electroluminescence data in this case.

8. It is described that the emission of the PTCDA probe is more intense than the emission of plasmon, but this is not clear because the emission spectrum of plasmon is set to 1/20.

The reviewer is correct in noting that our formulation misleadingly suggests an increased absolute value of collected photons, potentially integrated over a large spectral range. Although it has been observed in some cases, this is not a general behavior valid for all functionalized tips.

Action: We removed the misleading formulation.

9. Traces during the approach should also be shown in Fig. 1a. What is the applied voltage for Fig. 1a?

In Figure 1a, the last steps of the approach trace appear as a dashed line. During the approach and the first retraction step (till 2.15 nm), a voltage of 20 mV is used. For larger separation, the current could no longer be measured at this low voltage, this is why for $z > 2.15$ nm a larger voltage is used (2.5 V). This change of experimental conditions was mentioned in the caption, but the comment of the reviewer indicates that this aspect should be better described. In the amended manuscript we have modified the caption of Fig. 1 to make this point clear.

Action: The caption has been modified accordingly.

Reviewer 3: I co-reviewed this manuscript with one of the reviewers who provided the listed reports. This is part of the Nature Communications initiative to facilitate training in peer review and to provide appropriate recognition for Early Career Researchers who co-review manuscripts.

We thank the reviewer for their contribution to the peer-review of our manuscript.

Reviewer 4: The manuscript entitled "Fluorescent single-molecule STM probe" by N. Friedrich et al. reports on the functionalization with a 3,4,9,10-perylenetetracarboxylic-dianhydride (PTCDA) molecule of the tip of a low temperature scanning tunneling microscope (STM) acting as a fluorescent probe sensitive to the local electromagnetic and electrostatic fields of samples. The authors demonstrate in a series of scanning tunneling microscopy luminescence (STML) experiments supported by quantum chemical and quasi-electrostatic calculations the variation of the fluorescence of the molecular tip as a function of the targeted substrates. They compare STML spectra of the lifted PTCDA facing the pristine Ag(111) surface and NaCl/Ag(111) with the flat lying PTCDA molecule adsorbed on NaCl/Ag(111). Interestingly, the full width at half maximum (FWHM) of the emission peak is larger using the PTCDA tip in comparison to the flat-lying PTCDA molecule on NaCl using a Ag tip. This demonstrates that the molecule fluorescence is preserved when on the tip due to a reduced charge-transfer rate between the molecule and the tip. They also relate the FWHM to the molecular geometry inducing a dipole close to collinear with the plasmonic electric field in the gap. Last, the emission spectra on top of 2 ML NaCl is also red-shifted by as compared to the bare Ag(111), indicating that the PTCDA tip emission is sensitive to its local dielectric environment. Last, the authors explain using numerical calculations and models the absence of quenching of the suspended molecule's emission. To my opinion, the manuscript is well-written. The experimental data are of very high-quality and supported by advanced numerical calculations. The conclusion of the work is supported by both experimental and numerical data. It is a nice piece of work.

Overall, these experiments follow the general trend in low-temperature scanning probe microscopy, which focus on terminating the apex of an atomic force microscope or an STM with different molecules (CO, PTCDA, nickelocene etc.) to access new insights into the structural, electrostatic or magnetic properties of samples. By demonstrating the high spectral sensitivity of such fluorescent probe to its local environment, the authors shed new lights into Thus, I think this work will attract a broad attention not only for the technical aspects related to the STM/AFM techniques (single-molecule manipulation, STML spectroscopy of single-molecule tips) but also, and maybe more importantly, the physics potentially accessible with such techniques. Therefore, I recommend the publication of the present manuscript in Nature Communication after minor revisions.

We thank the reviewer for the detailed comments, and recommendation to publish the

present manuscript in Nature Communications after minor revisions.

Here is my only comment: 1- "Our data also demonstrate that the fluorescent properties of the molecular probe are sensitive to their electromagnetic and electrostatic environment. [...] Further work will be devoted to characterizing the spatial resolution that is achievable with this atomic-scale sensor that could be used, in the close future, in resonant energy transfer microscopy experiments having ultimate lateral and axial precision."

My feeling is that the authors should discuss the term "probe" to avoid confusions. The luminescence of the PTCDA probe is unambiguously sensitive to its local electrostatic environment as demonstrated in the manuscript by varying the sample, the bias voltage or the tip-sample distance. However, combining STM and fluorescent probe such as in the title suggests the combination of both the atomic scale resolution of the STM technique with the high spectral sensitivity of the PTCDA fluorescent tip. My only concerns is that the spatial resolution of the PTCDA tip along the X and Y directions is not demonstrated in the present work. For instance, it would be interesting that the authors provide a cross-section of STML spectra with the PTCDA tip across an NaCl island.

Again, we thank the reviewer for highlighting that our probe is "unambiguously sensitive to its local electrostatic environment". We also agree that the original version of the manuscript would be strengthened by a demonstration of the sensitivity of the probe in x, y direction. Whereas the acquisition of a full STM image together with spectral acquisition with a PTCDA-tip (as requested by reviewer 2) is beyond reach (such an image requires a very long acquisition time and extreme tip stability) we were able to record the STML spectra for each pixel of a line-scan for a PTCDA-tip that is progressively approached to an atomic-scale defect located on a NaCl bilayer. Results of this line-scan are shown here (Fig. R5), and in the amended manuscript, where we added a new figure (Fig. 3). A small, but clearly observable shift of the spectral line appears in the proximity of the defect. The unknown nature of the defect, as well as the limited number of acquired data points, prevent a full quantitative analysis of the effect, but this new set of data perfectly illustrates the potential of the PTCDA-tip that is clearly sensitive to atomic-scale variation of its environment. We hope that this set of data can lift the concern of the reviewer.

Action: We have added a new figure to the main text (Fig. 3) and some related description and discussion text (essentially in page 4).

Similar to scanning quantum dot microscopy (SQDM) (Reference 3), the sensitivity of the PTCDA fluorescent probe is related to the electric surface potentials, thus it might be subject to the same lateral resolution. Could the author discuss that? Do they expect submolecular resolution when performing STML maps on molecules with PTCDA tips such as Reference 25? I believe a small discussion along this line would be valuable for the manuscript.

The new data set reveal spectral changes for nanometer variations of the molecular position, though we feel that these data do not allow discussing the exact lateral resolution of our approach. Overall, more than being more or less resolved than other similar scanning probe approaches, we believe that the main perspective of molecular fluorescent tips is their ability to probe coherent dipole-dipole interactions or resonant energy transfers arising between two molecules, a unique capability that they share with no other functionalized tip.

REVIEWER COMMENTS

Reviewer 1: The authors have addressed all questions raised. The manuscript is recommended for publication.

We thank the reviewer for their feedback and appreciate that they recommend our manuscript for publication.

Reviewer 2: I appreciate that the authors considered our comments. The additional experiment around an atomic point defect on the NaCl surface (Fig. 3) is very interesting and seem containing important information. However, our assessment of this study has not been changed much. It examined the luminescence of molecules adsorbed on the probe. Demonstrations as new probes are primitive and are not well explained. Therefore, we cannot recommend publication of this manuscript in the current form. This is mainly because the fundamental question, "What can be measured with this probe?", has not been answered yet. We strongly recommend the authors to reconsider the following points for improving the manuscript.

We thank the reviewer for their report and for recognizing the importance of the new sets of data and simulations that were collected and performed following their requests in the previous report. We understand that, despite these efforts, the reviewer still found our results too "primitive" to demonstrate that the PTCDA-tip acts as a probe. They also feel that we do not explain well enough what can be measured with this probe. Below we provide answers to the detailed questions of the reviewer as well as arguments aiming to clarify our claims.

1. It is obvious, without need for discussion, that molecules change their properties depending on the environment or fields. Therefore, while the fact that a molecule adsorbed on the metal tip shows luminescence is in itself a surprise, it is expected that the resonance properties of the molecule-on-tip change depending on the environment. That's the reason of our assessment that this study examined the luminescence of molecules adsorbed on the probe. If the authors claim that the molecular tip can be used as a new probe, it is very necessary to demonstrate to measure something. Simply showing the property change is not measuring a quantity and is not enough for the claim.

Here, we must confess having difficulties to follow the reviewer's comment. Recognizing "that the resonance properties of the molecule-on-tip is affected by the environment" implicitly means that one can gather information on this environment by recording these changes. Conceptually, it therefore seems that the reviewer agrees with us. Their problem seems to be that we do not go far enough in the analysis of what we probe in this paper.

First, we want to emphasise that our paper should be viewed as a proof-of-principle that a molecular-tip obtained by STM manipulation can be used as a molecular probe. This was far from obvious, as it was believed that the direct molecule-tip contact would quench molecular light emission. Second, we believe that we are going far beyond "simply showing the property changes". We demonstrate experimentally and theoretically that the spectral width is assigned to changes in the electromagnetic environment of the molecule, something that the reviewer 2 does not acknowledge in their reviews. Second, the shift of the peak is shown to be dominated by changes in the electrostatic environment. We based this assertion on the voltage dependency of this feature (Fig 2d) and confirmed it by probing different surfaces (bare silver or NaCl bilayer). As requested by the reviewer, we eventually demonstrated that this parameter is sensitive to atomic-scale adsorbates. Altogether, these observations go far beyond "simply showing the property change" and cannot be resumed by writing that we "examined the luminescence of molecules adsorbed on the probe".

2. The authors discussed the physical origin of 21 meV shift in peak energy in the reply and in the revised manuscript. In the first manuscript it was ascribed to difference in local dielectric environment. Now after revision, it is explained by electrostatic environment (Stark effect). In our opinion, dielectric environment and electrostatic environment are very different. The material surrounding the molecule respond dielectrically to the dynamic movement of charge within the molecule in the excited state. If the material surrounding the molecule changes, its response will change, and thus the molecular resonance frequency is expected to be modulated. This is the effect of the dielectric environment, which is expected to result in a shift in peak energy when the molecular tip is on Ag and NaCl. Electrostatic field effect (Stark effect) is a shift in resonance energy due to an electrostatic field. The authors have changed their explanation, suggesting that not enough thought has been given to the key result, the energy shift. Now in the revised manuscript, the dielectric environment effect is completely ignored. As I mentioned in the previous comment, there are THREE, not two, possible causes for the peak energy shift; the dielectric environment, Lamb shift, and Stark effects. It is necessary to discuss the impact of all effects on the 21 meV energy shift, rather than ignoring them without rational reason. We would not recommend publication unless the physical origins of the energy shift are adequately discussed.

We thank the reviewer for their comment, however, we feel that it arises from a semantic problem: when using the term "dielectric environment" we intended to say that it would

shape both the electrostatic and electrodynamic properties of the system. Indeed, the role of a dielectric environment in electrostatics is ubiquitous and we therefore do not agree that the term "dielectric environment" should exclusively apply to dynamical phenomena (such as the Lamb shift) or exclusively to solvatochromism (as we believe the reviewer suggests). The role of a dielectric layer on the change of the work-function of metal surfaces is a well known phenomenon which we reflect in our interpretation. We therefore cannot agree that we changed our interpretation; on the contrary, in our revised version we have provided a more concrete plausible explanation of the observed effect and put more thought into the interpretation of a rather complex phenomenon. We eventually concluded that the observed spectral shift is dominated by changes in the electrostatic field of the PTCDA-tip environment. A more quantitative analysis of this phenomenon is certainly possible but, in our opinion, is far beyond the scope of our proof-of-principle study.

3. The additional experiment around an atomic point defect on the NaCl surface (Fig. 3) is very interesting. However, the results are not discussed at all, unfortunately. If the authors cannot discuss the results because the defects are unknown, then the results are not worthy of publication. At the very least, they should measure something known. Could that defect be a Cl vacancy in NaCl? Cl vacancies have been investigated in previous STM studies and can be created on NaCl films with an STM tip, so I think they could be considered a sample with well-understood properties. Since these results are an important demonstration, they should be revised and included in the paper.

In Fig. 2 we demonstrated that the peak position of our probe depends on the electrostatic environment of the molecular tip. The purpose of Fig. 3, besides satisfying the former requests of ref. 2 and 4, is that the peak position is sensitive enough to detect the presence of atomic-scale protrusions. We believe that Figure 3 satisfies this purpose. We understand that the referee would like to see some additional data on a more characterized defect. We agree with them that this is an interesting perspective for future works more focused on calibrating our fluorescent probe or directly focusing on the properties of the defect. Both aspects are, however, beyond the scope of this proof-of-principle study.

4. The data in Figure 3 are very interesting. The red shift is larger at a little distance from the molecule than at the closest point to the molecule. Why is this? Also, what is the

orientation of PTCDA in this case? Since the molecule is anisotropic, the orientation of the molecule must be known in order to interpret these results.

We also believe that the detailed response of the probe are affected by the orientation of the molecule at the tip. Characterizing this in detail would require recording optical spectra for each pixels of a 2D map around the atomic-scale defects, as well as calculating theoretical maps reproducing the shift for different orientations of the molecule on the tip. Whereas these are indeed interesting suggestions for future works, they also constitute very challenging experiments and time-costly simulations, and we do not feel that it is reasonable to request them in the frame of this proof-of-principle study.

5. Was the STM image in Fig. 3b measured with the molecular tip or metal tip?

Figure 3b was recorded with a metal tip before the attachment of a PTCDA to the tip apex. We have clarified this point in the manuscript.

Action: Added "acquired with a metal tip" to the caption of the Figure.

6. In response to my previous comment "If the authors claim that the molecule adsorbed on the probe is an STM probe, as the title suggests, then it is highly desirable to show STM images acquired with that probe.", they responded "We yet did not succeed in acquiring a full STM image together with spectral acquisition with a PTCDA-tip." I am not asking for something complicated. I am simply saying that STM images obtained with a PTCDA-tip should be shown. If the author refers to a PTCDA-tip as an STM tip, it is required.

We apologize for misunderstanding the former comment of the reviewer. Figure R1 shows an image acquired with a PTCDA-tip on top of a circular cluster of silver adatoms.

Action: The Section S1.E, including Figure S4, has been added to the SI of the manuscript.

FIG. R1: Left: Constant current STM image ($V = -2.5$ V, $I = 5$ pA) recorded with a clean metal tip showing the clean Ag(111) surface, partially covered with 2ML NaCl and a random distribution of adsorbed PTCDA molecules. A prominent circular cluster of Ag adatoms is protruding significantly from the Ag(111) surface. Right: After functionalizing the metal tip with a PTCDA molecule the appearance of the adatom cluster changes drastically. The adatom cluster appears as two bright lobes in a constant height STM image ($V = 2.5$ V) recorded with the tip retracted to $z \approx 2.4$ nm. The image resembles closely a constant height STM image recorded above an upright standing PTCDA molecule [Esat *et al.*, Physical Review Research **5**, 033200 (2023)] and, therefore, confirms the successful functionalization of the tip.

Reviewer 3: I co-reviewed this manuscript with one of the reviewers who provided the listed reports. This is part of the Nature Communications initiative to facilitate training in peer review and to provide appropriate recognition for Early Career Researchers who co-review manuscripts.

We thank the reviewer for their contribution to the peer-review of our manuscript.

Reviewer 4: The authors have addressed the comments and corrections satisfactorily.

We thank the reviewer for their feedback and appreciate that they recommend our manuscript for publication (see the first report).

REVIEWER COMMENTS

Reviewer 2: We are disappointed in the authors' decision not to discuss one of the most important results of the paper, the 21 meV energy shift between on Ag and on NaCl, just by saying it is beyond the scope of the paper. I believe that discussing the physical origins of the energy shift is not beyond the scope of the paper, but rather should be the core of the paper. Because without that discussion, one cannot conclude what this probe can measure. We cannot recommend publication of this manuscript in the current form. This is mainly because the fundamental question, "What can be measured with this probe?", has not been answered yet. We strongly recommend the authors to reconsider the following points for improving the manuscript.

1. Since the authors do not seem to be aware of the well-known phenomenon of dielectric-induced emission peak shifts, I list some examples of papers dealing with the effect.

[1] F. J. R. Costa, T. G.-L. Brito, I. D. Barcelos, and L. F. Zagonel, Impacts of Dielectric Screening on the Luminescence of Monolayer WSe₂, *Nanotechnology* 34, 385703 (2023).

[2] S. Borghardt, J.-S. Tu, F. Winkler, J. Schubert, W. Zander, K. Leosson, and B. E. Kardynal, Engineering of Optical and Electronic Band Gaps in Transition Metal Dichalcogenide Monolayers through External Dielectric Screening, *Phys. Rev. Mater.* 1, 054001 (2017).

[3] M. Florian, M. Hartmann, A. Steinhoff, J. Klein, A. W. Holleitner, J. J. Finley, T. O. Wehling, M. Kaniber, and C. Gies, The Dielectric Impact of Layer Distances on Exciton and Trion Binding Energies in van Der Waals Heterostructures, *Nano Lett.* 18, 2725 (2018).

It is well known that the presence of a dielectric material in close proximity to the emitting material can shift the emission peak by several 10 meV. I believe this effect cannot be ignored in the explanation of the 21 meV energy shift, since the materials surrounding the molecule are different on Ag or on NaCl.

Now in the manuscript, the dielectric screening effect from the surrounding is completely ignored. As I mention continuously in the comment, there are three, not two, possible causes for the peak energy shift; the dielectric environment, Lamb shift, and Stark effects. It is necessary to discuss the impact of all effects on the 21 meV energy shift, rather than ignoring them without rational reason. We would not recommend publication unless the physical origins of the energy shift are adequately discussed.

In our previous response (as well as in the manuscript) we discussed the microscopic origin of the shift of the emission-line energy and pointed out that it is consistent with the Stark shift induced by the change in the work function of the substrate. Nevertheless, we as well wish to unravel the microscopic mechanisms at play in our experimental setup and we propose a more in-depth discussion to address the reviewers comment. We are well aware of the general effect that the dielectric environment can have on the energy of excitonic emission and we have never questioned it (in fact we have discussed the dynamical "Lamb shift" in our related work, see [Rosławska et al., Phys. Rev. X, 12, 011012 (2022)]). We tried to discuss the microscopic mechanisms that may lead to the (vaguely defined) "dielectric effect". This "dielectric effect" (in chemistry discussed under the term solvatochromism - a term more appropriate to our molecular setup, see e.g. [Cammi et al., J. Chem. Phys., 122, 104513 (2005); Corni et al., J. Chem. Phys., 123, 135512 (2005); Coane et al., Commun. Chem., 7(1), 32 (2024)]) contains a number of microscopic phenomena including static dielectric screening (effect of the presence of static charges and their reorganization) and dynamical screening (also including the photonic Lamb shift).

We appreciate that the reviewer included the references discussing the effect of a dielectric on *Wannier* excitons in 2D materials to clarify their point. We do not question the existence of the effect of a dielectric discussed in the references. Indeed, both static and dynamical screening play a role in general. We would like to point out that there is a dramatic difference between how Wannier excitons are described and how one usually describes molecular excitations (more similar to Frenkel excitons). The differences not only appear in the terminology and methodology used to discuss the related phenomena, but potentially also affect the relative magnitude of such phenomena depending on the character of the excitation and the exact geometry of the environment.

Below we discuss well-defined concrete microscopic phenomena that can be associated with the broadly defined "dielectric effect": the Lamb shift (dynamical renormalization of the molecular excitation energy in response to the environmental *dynamical* screening; see e.g. [Rosławska et al., Phys. Rev. X, 12, 011012 (2022), Coane et al., Commun. Chem., 7(1), 32 (2024); Zhang et al., Nat. Commun., 8, 15225 (2017)]), the static Stark shift (the effect of static electric fields emerging from the environment independently of the presence of the molecule), and the *static* dielectric screening (the effect of static polarization of charges in the environment induced by the presence of the charged molecule).

1) We have estimated the value of the Lamb shift that would result from the inclusion of a thin insulating layer of NaCl. The Lamb shift can be defined as $\hbar E_{\text{Lamb}} = \text{Re}\{\int \rho \phi d\mathbf{r}\}$, where ρ is the transition density of the molecule and ϕ is the dynamical potential induced by the dielectric environment [Rosławska et al., Phys. Rev. X, 12, 011012 (2022)]. Due to the low refractive index of NaCl at optical frequencies (we stress here the difference between the static and dynamical screening), the effect of including the layer primarily manifests itself as an increased distance of the molecule from the Ag substrate. We have therefore evaluated the Lamb shift for a range of molecule-substrate separations (distance of the center of the molecule from the interface). The data is offset such that the value of the Lamb shift is zero at $z = 1.05$ nm, corresponding to the situation with the molecule suspended above the pristine Ag surface. To that end we used (i) an approximation where the molecule is treated as a point dipole positioned above the Ag interface (black diamonds in Fig. R1) and the tip is not considered, and (ii) a more complete model where both the tip and the substrate are present and the molecule is represented by its dynamically oscillating transition density (red diamonds). In the latter case we maintained the length of the tip which resulted in a shift of its plasmon resonance. We have therefore evaluated the Lamb shift at the energy corresponding to this plasmon resonance. In both cases we see that the Lamb shift *variation*, reaching units of meV (up to 2 meV considering 0.5 nm as the geometrical height of the NaCl layer), is not sufficient to explain the ~ 20 meV shift observed experimentally that would correspond to the difference between the Lamb shift evaluated for a smaller distance (closer to the Ag surface) and a value evaluated at a larger distance (distance to the Ag surface offset by the presence of NaCl). Moreover, we see that the Lamb effect would lead to a blue shift of the molecular resonance when the molecule-functionalized tip is moved from the clean Ag surface to the NaCl island, *i.e. to a behaviour qualitatively opposite to that observed experimentally*.

We remark in passing that it is not clear to us whether this important dynamical effect is included in ref. [M. Florian, et al., Nano Lett. 18, 2725 (2018)] provided by the reviewer since in ref. [M. Florian, et al., Nano Lett. 18, 2725 (2018)] it seems that only the static response of the dielectric is considered.

2) The *static* Stark effect is caused by the action of the static electric field present in the environment on the difference Δp between the static dipole moment of the molecule in the ground and excited state, respectively. In the experiment, the effect of molecule

FIG. R1: Values of the Lamb shift calculated as a function of the distance z of the center of the molecule from the interface. The Lamb shift was offset such that it is zero for the distance of $z=1.05$ nm. The calculation was done using a point-dipole approximation for the molecule without considering the tip (black diamonds), and considering the transition density and including the tip (red diamonds). In the latter case the tip-molecule distance was held constant and the value of the Lamb shift was evaluated at the resonance of the plasmon to avoid spurious effects due to the fact that the plasmon resonance is shifting with increasing z .

symmetry breaking due to its adsorption on the tip is expected to induce a static dipole moment in the molecule although the molecular symmetry in vacuum would suggest the absence of the static dipole moment. Δp can be estimated using the data derived from the Stark shift experiment (Fig. 2e). We estimate $\Delta p \equiv c_S = 0.06 e \cdot \text{nm}$ - as discussed in the manuscript. The static Stark shift is an important first-order phenomenon and we can estimate it experimentally via the bias and distance dependent measurements presented in the main text. The estimated 30 meV shift is consistent with the experimentally observed 21 meV shift of the emission line that can be associated with the voltage drop in the gap induced by the presence of the NaCl layer (associated with the change of the substrate work function). The detailed discussion of this non-trivial effect can be found in the main text and our previous answers to the reviewers.

3) We next address the effect of the induced *static* screening charges on the excitation energy, i.e. the effect explicitly discussed in ref. [M. Florian, et al., Nano Lett. 18, 2725 (2018)] evoked by the reviewer. In the leading order, it would result from the interaction of Δp with the static reorganization of charges in the dielectric caused by the static charge of

the molecule (since the molecule is negatively charged in our experiment). As the molecule is placed above an interface with a medium having a large *static* dielectric constant (both Ag and NaCl perfectly screen the static charge), the difference of the energy shift between the molecule-on-NaCl and molecule-on-Ag configuration comes from the geometrical distance z of the molecule from the substrate (and its variation Δz between the two configurations). We estimate the value of this induced shift of molecular emission line energy by considering the interaction between the charge in the dielectric, induced by the molecular interaction, with the static dipole moment of the molecule, as:

$$\Delta E_{\text{ind}} = \frac{ec_S}{8\pi\epsilon_0 z^3} \Delta z. \quad (\text{R1})$$

Where ϵ_0 is the vacuum permittivity and e the elementary charge. We insert $z = 1$ nm for the effective distance of the center of the molecule to the interface and $\Delta z = -0.05$ nm for the variation of it whether NaCl is present or not. The latter is estimated from the experimentally recorded position of the tip and the thickness of 2ML of NaCl. Using these values we estimate that the difference between the shift induced on NaCl and on Ag is about $\Delta E_{\text{ind}} \approx -2.2$ meV. That is, the effect does not explain the ~ 20 meV shift observed experimentally as it is an order of magnitude smaller than the experimental observation.

We thus conclude that it is likely the static Stark shift that explains the experimentally observed red shift of the molecular resonance when the molecule-functionalized tip is moved from the clean Ag surface to the NaCl islands as it is an order of magnitude larger than the remaining two effects considered. Moreover the Lamb shift would lead to a qualitatively different blue shift compared to the observed red shift.

Action: We added a supplementary section S2.I where we discuss the effects of the dielectric environment. In the main text we now say: "For completeness, in Section S2.I of the SI we discuss the possible role of other effects including the static and dynamic (Lamb shift) screening by the dielectric on the line shift. We conclude that the Stark shift due to the change in the substrate work function is one order of magnitude stronger than the other effects considered. Moreover, the Lamb shift would have the qualitatively opposite effect of blue shifting the resonance of the molecule on NaCl. Overall, this suggests that the peak shift observed in our data is dominated by the Stark effect."

2. Since the authors seem unwilling to explain the newly added data (Fig. 3), we strongly recommend that the data be excluded from the paper. If the data is measured

without knowing what kind of substance or physical quantity is being measured, what is the meaning of the data?

I am not making the comment for my own satisfaction. I am giving advice to improve the paper.

Here is my comment in the first round, "If the peak shift reflects the dielectric environment as the authors claim, one would expect it to vary with position on the NaCl film. It is significant to show that data to demonstrate the usage as a new kind of probe."

This comment points out that it is important to show nanoscale probe position dependence for claiming usage as a STM probe. Of course, the comment assumed that if the results are to be shown, the meaning should be explained.

As mentioned, we did our best to explain the extremely intricate effects that may be at play in this state-of-the-art experiment. The full explanation of all the effects potentially observed in the paper would probably close several fields of research in chemistry and solid state physics (as proven by the literature provided by the reviewer, among others) and therefore cannot be reasonably expected of us.

We included the data in Fig. 3 upon the request of the reviewers. Reviewers 1 and 4 seemed to have been satisfied with it. Faced with the contradiction of opinion we therefore prefer to include the data and, thus, increase the value and transparency of our manuscript.

We also believe that by including these new data we have demonstrated the probing capability of our technique, exactly as requested by the reviewer in the first round of reviews.

Reviewer 3 (Remarks to the Author): I co-reviewed this manuscript with one of the reviewers who provided the listed reports. This is part of the Nature Communications initiative to facilitate training in peer review and to provide appropriate recognition for Early Career Researchers who co-review manuscripts.

We thank the reviewer for contributing to the discussion.

REVIEWER COMMENTS

Reviewer #2 (Remarks to the Author): 4th Review of "Fluorescent single-molecule STM probe" by Friedrich et al.

The authors have newly discussed the screening effect of the NaCl film in the optical frequency domain, and I appreciate them for improving the quality of the paper by providing a more detailed description of the observed phenomena. This fluorescent probe is inherently sensitive to both the electrostatic field and the electric field in the optical frequency domain. In other words, the energy shift of the emission peak is caused by both the electrostatic field and the optical frequency electric field. Compared to the previous manuscript, which mainly discussed only the Stark effect, the current version describes the physical phenomena more accurately. I believe this description is crucial for understanding the characteristics of the new probe.

We are happy to read that the referee appreciates our efforts. However, we demonstrated the opposite of the referee's conclusion considering various contributions, and definitively and extensively ascribe the peak shift observed in Figure 2a to the sole Stark effect, despite the tip being sensitive to plasmonic effects, too (see the line width analysis in Figure 2). The plasmonic coupling to the environment does not induce strong shifts of the central emission line. Rather, the width of the peak is sensitive to the plasmonic environment, as shown, e.g., in Figure 2b making it potentially possible to simultaneously measure the two effects independently using the same probe.

There have already been studies using STM probes with PTCDA molecules adsorbed at the tip to probe electrostatic fields, electrostatic potentials, and magnetic fields.

[1] Wagner, C., Green, M. F. B., Leinen, P., Deilmann, T., Krüger, P., Rohlfing, M., Temirov, R., and Tautz, F. S. (2015). Scanning Quantum Dot Microscopy. *Physical Review Letters*, 115(2), 026101. <https://doi.org/10.1103/PhysRevLett.115.026101>

[2] Wagner, C., Green, Matthew. F. B., Maiworm, M., Leinen, P., Esat, T., Ferri, N., Friedrich, N., Findeisen, R., Tkatchenko, A., Temirov, R., and Tautz, F. S. (2019). Quantitative imaging of electric surface potentials with single-atom sensitivity. *Nature Materials*, 18(8), 853-859. <https://doi.org/10.1038/s41563-019-0382-8>

[3] Esat, T., Borodin, D., Oh, J., Heinrich, A. J., Tautz, F. S., Bae, Y., and Temirov, R. (2024). A quantum sensor for atomic-scale electric and magnetic fields. *Nature Nanotech-*

nology. <https://doi.org/10.1038/s41565-024-01724-z>

[4] Bolat, R., Guevara, J. M., Leinen, P., Knol, M., Arefi, H. H., Maiworm, M., Findeisen, R., Temirov, R., Hofmann, O. T., Maurer, R. J., Tautz, F. S., and Wagner, C. (2024). Electrostatic potentials of atomic nanostructures at metal surfaces quantified by scanning quantum dot microscopy. *Nature Communications*, 15(1), 2259. <https://doi.org/10.1038/s41467-024-46423-4>

If the authors claim, as in the current manuscript, that their fluorescent probe is primarily sensitive to the electrostatic field, it is highly desired to compare its performance and versatility with those of previous studies. This comparison would help readers appreciate the significance of developing this probe. Conversely, if the probe does not demonstrate superiority over prior research, it would not merit publication in *Nature Communications*.

We claimed, consistent with previous versions of the manuscript, that our probe is both sensitive to the electrostatic and the electromagnetic fields (the latter is shown in line width analysis in Fig 2). In our mind, these two aspects are, at least, of equal importance. This is still the claim in the current version of the manuscript. Note that none of the papers cited above report on probing the dynamic electromagnetic field of the sample - something that we clearly demonstrate in this proof-of-concept study. Furthermore, we would like to point out that the bright fluorescence of a molecular emitter directly bonded to a metallic electrode itself is a noteworthy scientific advance.

The suggested references [1] and [2] are discussed in our manuscript since its initial submission. We believe reference [3] is also relevant for our work and included it in our reference list. We did not discuss this reference in the initial submission, as reference [3] was *submitted and published* after our initial submission. Note that reference [3] even cites the arxiv version of our manuscript recognizing that our work reports on "new functionalities to image surface potentials".

Action: We added suggested reference [3] to our reference list (reference [8] in our manuscript).

On the other hand, the ability to probe electric fields in the optical frequency domain has not been reported, and I believe this aspect is highly valuable. I recommend adding a discussion and perspective on the development of a qualitatively different new probe.

We thank the referee for recognizing the fundamentally new aspect of our experiment. We

dedicate about the same space in the manuscript for the discussion of the sensitivity to the plasmonic (electromagnetic) and electrostatic environment. This reflects our understanding of an about equal importance of both aspects. However, to emphasize the qualitatively new aspect of the probe we adjusted the wording in our conclusion.

Action: We modified the conclusion to now read: "While our data also demonstrate qualitatively that the fluorescent properties of the molecular probe are *simultaneously* sensitive to the *dynamical* electromagnetic, *i.e.*, *plasmonic*, and electrostatic environment, additional work will be required for a full quantitative description of all effects." [Changes emphasized]

As for Fig. 3, still I believe it should be excluded from the paper. I am not asking for a full explanation of everything. However, at the very least, I believe it is necessary to provide information on what physical quantities are being measured or what substances are being examined. At present, both of these pieces of information are insufficient. In the aforementioned prior studies [1-4] using similar PTCDA probes, the substances being measured are confirmed at the atomic level. The physical quantities being measured are also discussed. Therefore, what I am requesting is not unreasonable, but rather a standard level of detail expected when claiming the development of a new probe.

These data were explicitly demanded and appreciated by the two other referees. We believe that they serve as an adequate proof-of-concept of nanometre scale resolution and atomic-scale-defect sensitivity achievable with our presented probe. Therefore, we disagree with the demand of removing the data, in agreement with the two other referees.

In summary, if the authors continue to assert that the probe is only sensitive to the electrostatic field, they must demonstrate its advantages over previous studies. Therefore, in its current form, I cannot recommend the manuscript for publication.

In direct contradiction with the referee's comment, we continue to assert that the probe is not only sensitive to the electrostatic field, but (again) is also sensitive to the electromagnetic field by the Purcell effect. This is the clear and unique advantage with respect to all the previous studies.

Reviewer 3 (Remarks to the Author): I co-reviewed this manuscript with one of the

reviewers who provided the listed reports. This is part of the Nature Communications initiative to facilitate training in peer review and to provide appropriate recognition for Early Career Researchers who co-review manuscripts.

We thank the reviewer for contributing to the discussion.